# AN ALGEBRAIC APPROACH TO APPROXIMATELY EQUIVARIANT NETWORKS

## ABSTRACT

Equivariant neural networks incorporate symmetries through group actions, embedding them as an inductive bias to improve performance. Prominent methods learn an equivariant action on the latent space, or design architectures that are equivariant by construction. These approaches often deliver strong empirical results but can involve architecture-specific constraints, large parameter counts, and high computational cost. We challenge the paradigm of complex equivariant architectures with a parameter-free approach grounded in representation theory. We prove that for an equivariant encoder over a finite group, the latent space must almost surely contain one copy of the regular representation for each linearly independent data orbit, which we explore with a number of empirical studies. Leveraging this foundational algebraic insight, we impose the regular representation as an inductive bias via an auxiliary loss, adding no learnable parameters. Our extensive evaluation shows that this method matches or outperforms specialized models in several cases, even those for infinite groups. We further validate our choice of the regular representation through an ablation study, showing it consistently outperforms defining and trivial representation baselines.

## 1 INTRODUCTION

When we consider the problem of designing a neural network to solve a given task, we commonly observe the existence of a symmetry group $G$ that acts naturally on the training data.[1] We illustrate a generic architecture in Figure 1, which we interpret broadly: $E$ may be any sort of feature extractor, such as in an encoder or classifier; and $D$ may be any final component that produces outputs from latent representations, such as a classifier head or decoder. On the input and output sets, the actions $\alpha_{\mathcal{X}}$, $\alpha_{\mathcal{Y}}$ transform the corresponding data, which we may want to be respected by our neural network.

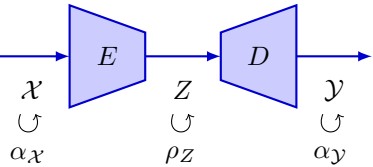

Figure 1: Generic architecture with input set $\mathcal{X}$, latent space $Z$ and output set $\mathcal{Y}$, carrying group actions $\alpha_{\mathcal{X}}$, $\alpha_{\mathcal{Y}}$ on the input and output spaces, and potentially a representation $\rho_Z$ on the latent space.

However, for certain tasks we can expect only *approximate equivariance*, where a transformation of the input vector corresponds inexactly, or nondeterministically, to a transformation of the outputs. This most general setup is typical of many real-world tasks, where we may encounter approximate scale-invariance or rotation-equivariance of turbulent dynamics (Holmes, 2012; Holl et al., 2020), and approximate reflection-invariance of pathologies in medical images (Yang et al., 2023).

A rich body of work in machine learning aims to learn a group representation $\rho_Z$ that acts linearly on the latent space, satisfying a suitable equivariance property. This can be attractive, as it may reduce a complex nonlinear action on the training set to an easily-computable linear function. Furthermore, this approach has been shown to yield improved performance for invariant, equivariant, or approximately equivariant tasks (see Section 2 for a brief survey). However, the performance benefits of many of

---

[1]We give a formal definition of a group action on a vector space (group representation) in Section 3.

these state-of-the-art methods often come at the cost of high model complexity, increased training times, and significantly elevated parameter counts compared to their non-equivariant counterparts.

Our research is guided by the following question: **for a known finite symmetry group, can we leverage a theoretically-principled understanding of the latent algebraic structure to achieve the benefits of (approximate) equivariance, without the parameter and architectural costs of current methods?** Our core theoretical contribution is a proof that for any equivariant encoder the latent space must contain the regular representation almost surely. Based on this finding, we propose a new, lightweight training regime: we fix the latent representation to be a multiple of the regular representation, and enforce this algebraic prior with an auxiliary loss. This approach yields strong performance on a variety of invariant, equivariant, and approximately-equivariant tasks. We summarize our main contributions as follows.

- We present a new lightweight method with no additional learnable parameters for training neural networks to solve invariant, equivariant and approximately-equivariant tasks, where a finite group acts on the training set with a known action.

- We provide a theoretical characterization of latent space representations under data augmentation and an equivariant encoder, showing that the regular representation must appear almost surely. Building on this insight, we empirically validate that neural networks tend to learn a linear action aligned with this structure.

- We show that our method is competitive with or exceeds state-of-the-art in a range of benchmarks, despite having only a single tunable hyper-parameter, and no additional learnable parameters, while alternative approaches typically have large learnable parameter demands (in some cases 5-20 times baseline) to achieve competitive performance.

## 2 RELATED WORK

A wide variety of methods have been developed to train neural networks to solve tasks in the presence of invariance, equivariance, or approximate equivariance. We give a brief summary here of those methods which are most relevant for our present work.

One of the most studied bodies of work derive from Convolutional Neural Networks (CNNs), which of course have strict translation invariance in their traditional form (LeCun & Bengio, 1998; Shorten & Khoshgoftaar, 2019). Cohen & Welling (2016) employ the framework of steerable functions (Hel-Or & Teo, 1998) to construct a rotation-equivariant Steerable CNN architecture (**SCNN**), which strictly respects both translation and rotation equivariance; this was later generalised to develop a theory of general E(2)-equivariant steerable CNNs (**E2CNN**), which allow the degree of equivariance to be controlled by explicit choices of irreducible representation of the symmetry group (Weiler & Cesa, 2019). Such a network can avoid learning redundant rotated copies of the same filters. A similar method is that of Mobius Convolutions (**MC**) (Mitchel et al., 2022). Wang et al. (2022) use steerable filters to obtain convolution layers with approximate translation symmetry and without rotation symmetry (**RSteer**), and with approximate translation and rotation symmetry (**RGroup**). These authors relax the strict weight tying of E2CNNs, replacing single kernels with weighted linear combinations of a kernel family, with coefficients that are not required to be rotation- or translation-invariant. A third approach named Probabilistic Steerable CNNs (**PSCNN**) was proposed recently by Veefkind & Cesa (2024), which allows SCNNs to determine the optimal equivariance strength at each layer as a learnable parameter. While equivariant architectures may allow reduced parameter counts due to weight-tying, in practice many of these architectures require considerable additional parameter counts to achieve competitive performance (see parameter counts in Section 6).

We also discuss a family of approaches which are not based around the CNN architecture. Residual Pathway Priors (**RPP**) (Finzi et al., 2021), is a model where each layer is doubled, yielding a first layer with strong inductive biases, and a second layer which is less constrained, with final output is obtained as the sum of these layers. Another architecture is Lift Expansion (**LIFT**), which factorizes the input space into equivariant and non-equivariant subspaces, and applies different architectures to each (Wang et al., 2021).

A number of previous studies have considered group representations on the latent space, sometimes governed via an equivariance term in the loss function. An early approach by (Welling & Cohen, 2014)

shows how geometrical transformations can be encoded on the latent space via SO(3) representations on the latent space, while (Worrall et al., 2017) demonstrate disentanglement phenomena with similar methods. Dupont et al. (2020) propose a parameter-free method to learn equivariant neural implicit representations for view synthesis; while similar to our method in some respects, such as fixing the latent representation, their work strongly leverages the defining representation of the infinite group $O(3)$, is limited to latent spaces with the same geometrical structure as the input space, and does not apply to arbitrary latent encodings. Jin et al. (2024) present a similar method which learns non-linear group actions on the latent space using additional learnable parameters, augmented by an optional attention mechanism. In Neural Isometries (**NIso**) (Mitchel et al., 2024), the authors propose to learn an action on the latent space via its eigenbasis; in contrast, in our model the group acts linearly on the latent space with a fixed representation, and with no additional parameters needed. Recent work of (Bökman et al., 2024) considers learned latent representations for a fixed group to solve certain geometrical tasks. Other approaches that do not require the symmetry group to be known beforehand include Neural Fourier Transforms (**NFT**) (Koyama et al., 2024), which seeks to learn a suitable latent space transformation, and other work (Shakerinava et al., 2022; Winter et al., 2024).

While our work builds on these approaches, our contribution is distinct: by assuming a known symmetry, we leverage representation theory to identify the regular representation as a theoretically-motivated latent structure, which enables our simple pipeline without additional learnable parameters. Although our approach requires fixing a group structure, our experimental results show that this approach can often achieve superior performance compared to models without this constraint, including models specifically adapted for continuous symmetries.

## 3 BACKGROUND ON GROUP REPRESENTATIONS

We review essential aspects of group representation theory for our work. We consider a finite group $G$ and work over a base field $\mathbb{K}$, assumed to be $\mathbb{R}$ or $\mathbb{C}$. The results presented are standard, for which we recommend canonical texts such as Fulton & Harris (2004) and James & Liebeck (2001). A glossary of notation and further background on group actions are available in Appendix B and C.

**Regular representation.** For the case of a finite group, the *regular representation* $\rho_{\mathrm{reg}}$ is defined as the linearisation of the action of $G$ on itself. Explicitly, we first define $\mathbb{K}[G]$ as having elements given by linear combinations of group elements $\sum_i c_i g_i$ weighted by coefficients $c_i \in \mathbb{K}$. Then $\rho_{\mathrm{reg}}$ is defined as a representation on $\mathbb{K}[G]$ as follows: $\rho_{\mathrm{reg}}(g)(\sum_i c_i g_i) = \sum_i c_i (g g_i)$. By construction we have $\dim(\rho_{\mathrm{reg}}) = |G|$, the size of the group. A representation $\rho$ on the vector space $\mathbb{K}^n$ is a *permutation representation* when for all $g \in G$, the matrix $\rho(g)$ is a permutation matrix. By construction, the regular representation is a permutation representation.

**Irreducibility.** Given vector spaces $V$ and $V'$ we can form their direct sum $V \oplus V'$, with elements which are ordered pairs of elements $(v, v')$ of $V$ and $V'$ respectively. Given a representation $\rho$ on $V$, and $\rho'$ on $V'$, we can form their *direct sum* $\rho \oplus \rho'$ acting on the vector space $V \oplus V'$, as defined as $(\rho \oplus \rho')(g)(v, v') := (\rho(g)(v), \rho'(g)(v'))$. For an integer $n$, we can similarly define the *n-fold multiple* of $\rho$, written $n \cdot \rho$, as $\rho \oplus \rho \oplus \cdots \oplus \rho$. If $\rho = \rho' \oplus \rho''$, we say that $\rho'$ and $\rho''$ are *subrepresentations* of $\rho$.

A representation is *irreducible*, also called an *irrep*, if it is not isomorphic to a direct sum of other representations, except for itself or the zero representation. A finite group has finitely many irreps up to isomorphism, and the regular representation is the direct sum of irreps, with each irrep taken with multiplicity given by its dimension. For example, the group $S_3$ has just the trivial (dim 1), sign (dim 2) and standard (dim 2) irreps (with the same for $D_3$ as they are isomorphic groups); and the cyclic group $C_n$ has $n$ irreducible representations (all dim 1) over $\mathbb{C}$, one for each $n$th root of unity.

**Orthogonality of representations.** For a fixed group $G$, we may ask whether a representation $\rho$ contains an irreducible representation $\rho'$ as a direct summand, and if so with what multiplicity. This can be determined using the formula for *inner product of representations*:

$$\langle \rho, \rho' \rangle = \tfrac{1}{|G|} \sum_{g \in G} \overline{\mathrm{Tr}(\rho(g))} \mathrm{Tr}(\rho'(g))$$

Given the knowledge of all irreducible representations of a finite group, this method allows us to determine their multiplicities as subrepresentations of $\rho$.

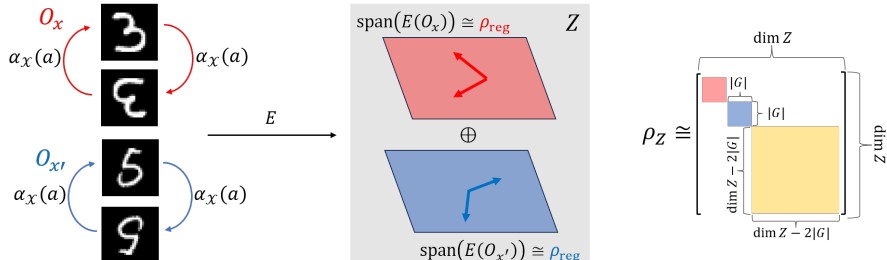

Figure 2: Illustration of our theory for an equivariant encoder $E$ and $G = C_2 = \{1, a\}$, with $\alpha_{\mathcal{X}}$ acting by horizontal flips. If $E(\mathcal{O}_x)$, $E(\mathcal{O}_{x'})$ are full rank and linearly independent, $Z$ must contain a separate copy of the regular representation $\rho_{\text{reg}}$ for each with probability 1.

## 4 IDENTIFYING OPTIMAL REPRESENTATIONS

We suppose a network architecture as illustrated in Figure 1 is given, with training elements $(x_i, y_i) \in \mathcal{X} \times \mathcal{Y}$, and task loss $L_{\text{task}}(D(E(x_i)), y_i)$. We now suppose a finite symmetry group $G$ is specified, which acts by fixed actions $\alpha_{\mathcal{X}}, \alpha_{\mathcal{Y}}$ on the input and output spaces respectively. We are interested to answer the following question: if we use additional learnable parameters to construct a third representation $\widehat{\rho}_Z$ of $G$ on the latent space $Z$, which we co-train alongside the parameters for $E, D$ with a suitable loss function, **what representation $\widehat{\rho}_Z$ does the model prefer to learn?** We first provide a theoretical analysis, which we then complement with an empirical exploration.

### 4.1 THE LATENT SPACE MUST CONTAIN THE REGULAR REPRESENTATION ALMOST SURELY

Adopting the notation above, we denote the $G$-orbit of a training sample $x \in \mathcal{X}$ as $\mathcal{O}_x := \{\alpha_{\mathcal{X}}(g)(x) \mid g \in G\}$. This contains all $G$-augmented versions of $x$, which we call the *data orbit of* $x$. We suppose $x$ is a single data sample chosen such that all augmented versions are distinct, i.e. such that $\alpha_{\mathcal{X}}(g)(x) = \alpha_{\mathcal{X}}(h)(x)$ implies $g = h$, which is typical for data augmentation. As a consequence $|\mathcal{O}_x| = |G|$, and we conclude that $G$ acts freely and transitively on $\mathcal{O}_x$ (nLab, 2024). We will be interested in the encodings $E_\theta(\mathcal{O}_x)$, where $E_\theta : \mathcal{X} \to Z$ is an encoder parameterized by $\theta \in \Theta \subseteq \mathbb{R}^p$, and with $\dim(Z) \geq |G|$. If $\dim(\text{Span}(E_\theta(\mathcal{O}_x))) = |G|$, then we say that $E_\theta(\mathcal{O}_x)$ is *full rank*, or otherwise *rank deficient*. We then show the following (proofs in Appendix E).

**Theorem 1** (Informal). *For an equivariant encoder $E_\theta$ and a training sample $x$, if $E_\theta(\mathcal{O}_x)$ is full rank, then the latent space contains a copy of the regular representation of $G$.*

**Theorem 2** (Informal). *If $E_\theta$ is also real analytic in its inputs and parameters, and trained by gradient descent, then for each training sample $x \in \mathcal{X}$, exactly one of the following holds:*

*(i) for all possible parameterisations $\theta \in \Theta$, the vectors $E_\theta(\mathcal{O}_x)$ are rank deficient.*

*(ii) with probability 1, the vectors $E_\theta(\mathcal{O}_x)$ are full rank, and hence the latent space contains the regular representation.*

Analyticity is discussed in Appendix E.1. We discuss the two cases in the statement of Theorem 2. Case *(i)* may arise in certain restricted cases—for example, if $E_\theta$ is $G$-invariant by construction— where no regular representation appears. However, for any training sample $x \in \mathcal{X}$, the two scenarios can be easily distinguished: sample $\theta \in \Theta$, then check rank deficiency of $E_\theta(\mathcal{O}_x)$. If rank deficiency holds, we are in case *(i)* with probability 1. Otherwise we are in case *(ii)* with probability 1. Furthermore, this principle extends across the training set, with each linearly independent full rank embedded orbit (Appendix E, Definition 4) contributing a separate copy of the regular representation.

> **Key theoretical insight:** To achieve encoder equivariance in the presence of data augmentation, a sufficiently large latent space must contain a separate copy of the regular representation for each linearly independent full rank embedded orbit. This is summarized in Figure 2.

The question remains how many copies of the regular representation one obtains in practice, and we investigate this with the following empirical studies.

## 4.2 Empirical exploration

For our empirical investigation, we conduct experiments with the following loss function:

$$L_{\text{opt}} = L_{\text{task}}\big(D(E(x_i)), y_i\big)$$

**Task loss.** Trains encoder and decoder on the supervised objective.

$$+ \lambda_t \, L_{\text{task}}\big(D(\widehat{\rho}_Z(g)E(x_i)), \, \alpha_{\mathcal{Y}}(g)(y_i)\big)$$

**Latent→Output Equivariance.** Encourages $D(\widehat{\rho}_Z(g)E(x))$ to match $\alpha_{\mathcal{Y}}(g)(y)$.

$$+ \lambda_e \, \text{MSE}\big(\widehat{\rho}_Z(g)E(x_i), \, E(\alpha_{\mathcal{X}}(g)(x_i))\big)$$

**Input→Latent Equivariance.** Encourages $\widehat{\rho}_Z(g)E(x)$ to match $E(\alpha_{\mathcal{X}}(g)x)$.

$$+ \lambda_a \big(\text{ALG}_{G,d} + \text{REG}_{G,d}\big)$$

**Algebra Loss.** Encourages algebraic properties for $\widehat{\rho}_Z$ to be a group representation.

We give additional insight into the algebra loss in Appendix D. Drawing insight from Theorems 1 and 2, for experiments involving analytic encoders, we expect to learn a representation $\widehat{\rho}_Z$ that contains copies of the regular representation. The number of copies is lower bounded by the number of linearly independent embedded data orbits, which must be empirically determined (details in Appendix F.1). In this section we describe a number of exploratory studies based on the MNIST (Deng, 2012), TMNIST (Magre & Brown, 2022) and CIFAR10 (Krizhevsky, 2009) datasets, for both autoencoder and classifier tasks, and for the groups $C_2$, $D_3$ and $C_4$. These show that for an analytic encoder $E$, when $\widehat{\rho}_Z$ is randomly initialized according to $\mathcal{N}(\mathbf{0}, \mathbf{1})$, the network prefers to learn a representation which consists *entirely* of linearly independent copies of the regular representation. Appendices F.4 and F.5 investigate alternative layer depths and initialization schemes, respectively.

**Non-analytic encoders.** While Theorem 1 applies to non-analytic encoders, Theorem 2 requires analyticity. In deep learning architectures most components are analytic (discussion in Appendix E.1), with the exception of some common activation functions such as ReLU, which are piecewise analytic. The Stone-Weierstrass theorem states that any continuous function can be arbitrarily well approximated on any bounded domain by an analytic function. We explore the representations learned for non-analytic encoders in Appendix F.3, where we re-run the exploratory experiments of this section with piecewise-analytic activations (ReLU), and show that the same conclusions hold: the network prefers to learn a representation which consists entirely of linearly independent copies of the regular representation.

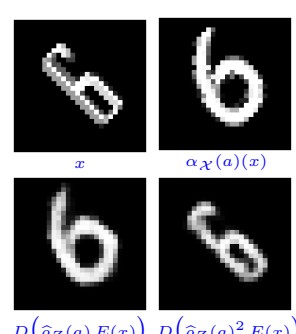

$x$      $\alpha_{\mathcal{X}}(a)(x)$

$D\big(\widehat{\rho}_Z(a)\,E(x)\big)$    $D\big(\widehat{\rho}_Z(a)^2\,E(x)\big)$

Figure 3: Visualisation of our learned encoder $E$, decoder $D$ and latent action $\widehat{\rho}_Z$ on input vector $x$ with $\alpha_{\mathcal{X}}$ swapping fonts. The algebraic loss correctly enforced $\widehat{\rho}_Z(a)^2 = I_d$.

### 4.2.1 TMNIST autoencoder, CNN architecture, $G = C_2$

For our first experiment we use the TMNIST dataset, of digits rendered in a variety of typefaces. We choose a subset of two typefaces only, producing 20 images, augmenting with 180° rotations. For our group we choose $G = C_2$ presented as $\{1, a \,|\, a^2 = 1\}$. Since this is an autoencoder we have $\mathcal{X} = \mathcal{Y}$, and we choose $\alpha_{\mathcal{X}} = \alpha_{\mathcal{Y}}$, with the nontrivial element $\alpha_{\mathcal{X}}(a) = \alpha_{\mathcal{Y}}(a)$ acting to flip the choice of font, with rotation and scaling left invariant. For the algebra loss component (*iv*) we choose $\text{ALG}_{C_2,d} = \text{MSE}(\widehat{\rho}_Z(a)^2, I_d)$ where $d = \dim(Z) = 8$.

Table 1: Left (Right) TMNIST (MNIST) analytic autoencoder task, learned representations of $C_2$ ($D_3$) on the latent space. $Z$ is taken as the middle layer, which carries the output of the encoder.

| | Irrep. counts | | | | | | Irrep. counts | | | | | |
|---|---|---|---|---|---|---|---|---|---|---|---|---|
| Run | −1 | +1 | Alg. loss | Eq. loss | Orbs | Run | Triv | Sgn | Std | Alg. loss | Eq. loss | Orbs. |
| 1 | 3 | 5 | $9.9\times10^{-10}$ | $1.6\times10^{-3}$ | 3 | 1 | 2.98 | 3.1 | 5.98 | $1.1\times10^{-4}$ | $1.4\times10^{-2}$ | 3 |
| 2 | 4 | 4 | $1.1\times10^{-7}$ | $1.3\times10^{-3}$ | 3 | 2 | 3.03 | 2.98 | 6.01 | $5.8\times10^{-3}$ | $2.1\times10^{-3}$ | 3 |
| 3 | 3 | 5 | $7.4\times10^{-10}$ | $1.2\times10^{-4}$ | 4 | 3 | 3.1 | 2.97 | 5.70 | $1.6\times10^{-4}$ | $2.7\times10^{-2}$ | 3 |
| 4 | 4 | 4 | $2.3\times10^{-10}$ | $9.1\times10^{-5}$ | 4 | 4 | 2.96 | 3.15 | 5.75 | $2.3\times10^{-3}$ | $3.1\times10^{-3}$ | 3 |
| 5 | 4 | 4 | $2.9\times10^{-9}$ | $1.5\times10^{-3}$ | 3 | 5 | 2.91 | 2.99 | 6.02 | $7.5\times10^{-2}$ | $2.2\times10^{-2}$ | 3 |

Table 2: CIFAR classifier task with analytic encoder, representations of $C_4$ learned on latent space. $Z$ is taken as the main feature layer before the final classification head.

| | Irreducible counts | | | | | | |
|---|---|---|---|---|---|---|---|
| Run | $+1$ | $+i$ | $-1$ | $-i$ | Alg. loss | Eq. loss | Orbs. |
| 1 | 4 | 4 | 4 | 4 | $1.5 \times 10^{-4}$ | $1.8 \times 10^{-3}$ | 4 |
| 2 | 3 | 4 | 5 | 4 | $7.2 \times 10^{-5}$ | $1.9 \times 10^{-3}$ | 3 |
| 3 | 3 | 5 | 3 | 5 | $9.4 \times 10^{-5}$ | $1.6 \times 10^{-3}$ | 3 |
| 4 | 4 | 4 | 4 | 4 | $1.1 \times 10^{-4}$ | $1.9 \times 10^{-3}$ | 4 |
| 5 | 4 | 4 | 4 | 4 | $8.4 \times 10^{-5}$ | $1.9 \times 10^{-3}$ | 4 |

Table 1 shows our findings, with each run giving one row of the table, and Figure 3 shows a visualization. Low values in the algebra and equivariance loss columns reveal high-quality representations $\widehat{\rho}_Z$, which are strongly equivariant with respect to the representations $\alpha_{\mathcal{X}}, \alpha_{\mathcal{Y}}$. By mapping the eigenvalues of $\widehat{\rho}_Z(a)$ to the nearest value in $\{-1, +1\}$, we can determine the corresponding irreducible representation. For the group $C_2$ the regular representation contains one copy of the -1 and +1 representations, and see that the learned $\widehat{\rho}_Z$'s are close to a multiple of the regular representation. Furthermore, we report the number of linearly independent embedded orbits and, as expected, this corresponds to the number of copies of the regular representation found (Section 4.1).

### 4.2.2 MNIST AUTOENCODER, MLP ARCHITECTURE, $G = D_3$

For our second experiment we choose the MNIST dataset of handwritten digits, augmented by arbitrary rotations. We choose the group $G = D_3$, the group of symmetries of an equilateral triangle with the generators $r, s$ (120-degree rotation, flip) and the following presentation: $\{e, r, r^2, r^3, s, rs \mid r^3 = e, s^2 = e, rsrs = e\}$. We parameterize the linear maps $\widehat{\rho}_Z(r)$ and $\widehat{\rho}_Z(s)$ independently, and define the following algebra loss, where $d = \dim(Z) = 18$, and where summands correspond to constraints in the presentation: $\mathrm{ALG}_{D_3, d} = \mathrm{MSE}(\widehat{\rho}_Z(r)^3, \mathrm{I}_d) + \mathrm{MSE}(\widehat{\rho}_Z(s)^2, \mathrm{I}_d) + \mathrm{MSE}(\widehat{\rho}_Z(r)\widehat{\rho}_Z(s)\widehat{\rho}_Z(r)\widehat{\rho}_Z(s), \mathrm{I}_d)$.

For the nonabelian group $D_3$, we determined the learned representation's composition using orthogonality of characters (Section 3). The data in Table 1 confirms that the network learns a high-fidelity multiple of the regular representation, which contains the trivial, sign, and standard irreducible representations in the ratio 1:1:2. Consistent with the previous experiment, each linearly independent data orbit contributes one distinct copy of this representation. Furthermore, Figure 4 illustrates the eigenvalues of the generator $\widehat{\rho}_Z(r)$ dynamically clustering around the third roots of unity during training, despite an uneven initialization.

### 4.2.3 CIFAR10 CLASSIFIER, CNN ARCHITECTURE, $G = C_4$

This experiment uses the CIFAR10 image dataset (Krizhevsky, 2009). We choose the group $G = C_4$ of 90-degree rotations, with the algebraic loss function $\mathrm{ALG}_{C_4, d} = \mathrm{MSE}\left(\widehat{\rho}_Z(1)^4, \mathrm{I}_d\right)$, where $d = \dim(Z) = 16$. For $C_4$ the regular representation contains exactly one copy of the $+1, +i, -1$ and $-i$ representations, and Table 2 shows that the network learns a representation close to a multiple of the regular representation. Furthermore, each linearly independent embedded data orbit contributes a distinct copy of this representation.

Considering these three experiments together, we summarize the results of this section follows.

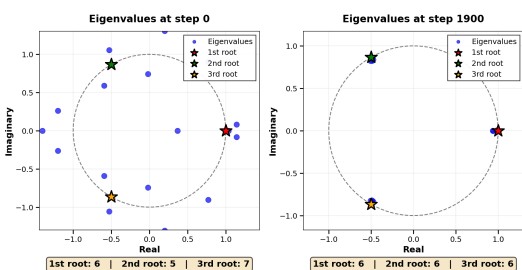

Figure 4: Eigenvalues of the real-valued matrix $\widehat{\rho}_{\mathcal{Z}}(r)$ at different training steps. Beneath each plot we show counts of eigenvalues nearest to each third root of unity.

**Key empirical insight:** To achieve encoder equivariance in the presence of data augmentation, the network prefers to learn a multiple of the regular representation on the latent space.

## 5 FIXING THE REGULAR REPRESENTATION

We present a novel parameter-free method to improve performance of neural networks to solve a variety of invariant, equivariant or approximately equivariant tasks, where a finite group $G$ acts on the input and output layers with representations $\rho_\mathcal{X}$ and $\rho_\mathcal{Y}$ respectively. Inspired by the theoretical and empirical results of Section 4, **instead of learning a representation on the latent space, we now fix $\rho_Z$ to be a multiple of the regular representation of** $G$. Specifically, we use $n$ copies where $n$ is the maximum number of representations allowed by dim $Z$. When $n|G| < \dim(Z)$, we pad by taking the direct sum with additional copies of the trivial representation, to ensure our representation on $Z$ has the correct dimension. Our proposed representation is therefore given by:

$$\rho_Z := n \cdot \rho_{\mathrm{reg}} + \max(\dim(Z) - n|G|, 0) \cdot \rho_{\mathrm{triv}} \tag{1}$$

When the latent space is geometrically structured, for example as a product of features and channels, we choose an isomorphic form of the regular representation that preserves this structure (examples are the SMOKE and SHREC experiment in Section 6). We then train according to the following objective function, where $(x_i, y_i) \in \mathcal{X} \times \mathcal{Y}$ is an element of the training set, $g \in G$ is a group element, and $L_{\mathrm{task}}(x_i, y_i)$ is the original task loss function:

$$\frac{1}{2} L_{\mathrm{task}}\big(D(E(x_i)), y_i\big) \qquad \text{Task loss}$$
$$+ \frac{1}{2} L_{\mathrm{task}}\big(D(E(\alpha_\mathcal{X}(g)x_i)),\ \alpha_\mathcal{Y}(g)y_i\big) \qquad \text{Task loss shifted by } g$$
$$+ \lambda\, \mathrm{MSE}\big(E(\alpha_\mathcal{X}(g)x_i),\ \rho_Z(g)\, E(x_i)\big) \qquad \text{Equivariance loss from input to latent}$$

When used in a training loop, we select $(x_i, y_i)$ and $g$ uniformly at random. Here $\lambda$ is a hyperparameter expressing the strength of the equivariance loss. We provide a sensitivity analysis for $\lambda$ in Appendix H, which shows that model performance is robust across a range of values. Our model has no additional learned parameters above baseline, since the representation $\rho_Z$ is now fixed. Our use of the $g$-shifted task loss means that our training dataset must be augmented by the action of $G$. This can be done either on-the-fly, or pre-computed to speed up training.

## 6 EXPERIMENTS

We benchmark our method against a variety of state-of-the-art methods for networks with approximate equivariance, considering four distinct tasks. We compare our results against the models SCNN, E2CNN, LIFT, RPP, RGroup, RSteer, PSCNN, NIso, NFT and MC, discussed in Section 2. All our experiments follow the setup of the original papers. Our method trains using a computational budget and wall-clock time at or below competing models. Since our model relies on data augmentation, we provide both augmented and unaugmented CNN baselines. Full technical details for all reported runs, including hyperparameter selection and a sensitivity analysis for our equivariance coupling strength $\lambda$, are reported in Appendix G and H. In the majority of cases our results are improved or comparable with state-of-the-art, while using fewer learnable parameters and a simpler architecture. We use Cohen's $d$-statistic to compute effect sizes (Hermann et al., 2024, p59), which shows that our model typically delivers very large performance improvements A discussion of these statistics can be found in Appendix G.1. For the selection of the layer $Z$ where the equivariance loss is imposed, for autoencoder tasks this is chosen as the output layer of the encoder, while for classification tasks we choose the layer before the final classifier head. As an ablation, we also report a comparison that replaces the regular representation in Equation 1 with the *defining representation*, the natural geometric action of the group by permutations (see Appendix C for a formal definition), and the trivial representation; these results further confirm optimality of the regular representation.

### 6.1 CLASSIFICATION TASK, DDMNIST, $G = C_2, C_4, D_4$

Following closely the procedure of (Veefkind & Cesa, 2024) for each of the chosen symmetry groups $C_2$, $C_4$ and $D_4$, we randomly and independently transform two MNIST images according to the group. Results are shown in Table 3. Because the transformations are local and independent, we apply our method using the product group. We also provide a comparison with the defining and trivial representations as an ablation study. While for the groups $C_2$ and $C_4$ the two representations are isomorphic, for $D_4$ they are not, with the regular representation being more performant; this provides further empirical evidence for the optimality of the regular representation. Except for SCNN, we

re-trained and re-evaluated all models. Further discussion and effect size analysis can be found in Appendix G.2. These statistics show a very large effect size for our model over the CNN baseline, and a large effect size for our model compared to the majority of results for other architectures.

Table 3: DDMNIST test accuracies. Mean over 3 runs; standard deviation in brackets. Parameter counts shown. Best result in each column is bold, second-best is underlined. For $C_2, C_4$ the defining representation is equivalent to the regular representation and so is omitted.

| Model | $C_4 \uparrow$ | #Params(M)$\downarrow$ | $C_2 \uparrow$ | #Params(M)$\downarrow$ | $D_4 \uparrow$ | #Params(M)$\downarrow$ |
|---|---|---|---|---|---|---|
| CNN | 0.907 (0.004) | **0.03** | 0.938 (0.006) | **0.03** | 0.800 (0.001) | **0.03** |
| SCNN | 0.484 (0.008) | 0.12 | 0.474 (0.003) | **0.03** | 0.431 (0.010) | 0.15 |
| Restriction | 0.914 (0.007) | 0.12 | 0.890 (0.007) | 0.33 | 0.837 (0.020) | 0.17 |
| RPP | 0.908 (0.022) | 0.79 | 0.903 (0.009) | 0.08 | 0.827 (0.020) | 1.73 |
| PSCNN | 0.909 (0.007) | 0.51 | 0.871 (0.016) | 0.04 | 0.842 (0.011) | 1.23 |
| Trivial rep | 0.874 (0.004) | **0.03** | 0.938 (0.007) | **0.03** | 0.819 (0.004) | **0.03** |
| Defining rep | – | | – | | 0.838 (0.010) | **0.03** |
| Ours (regular) | **0.915** (0.004) | **0.03** | **0.947** (0.004) | **0.03** | **0.868** (0.002) | **0.03** |

## 6.2 Classification Task, MedMNIST3D, $G = \mathrm{Sym}_{\mathrm{cube}}$

We test our method on the Organ, Synapse and Nodule subsets of the MedMNIST3D dataset, using the same setup as the original authors (Veefkind & Cesa, 2024). We apply the group $\mathrm{Sym}_{\mathrm{cube}}$ of orientation-preserving symmetries of the cube, which is isomorphic to the permutation group $S_4$. All results, except for ours and the augmented CNN, are imported from the original authors. Table 4 shows MedMNIST3D accuracies for different models and groups. For Nodule and Synapse, our method is comparable or outperforms other architectures, while having fewer parameters. The regular representation consistently outperforms the defining and trivial representations, providing further empirical evidence for its optimality. For the Organ dataset, canonical orientation is a key feature, and so the symmetry action conflicts with the task. This may explain our method's underperformance in this task (shared by the augmented CNN baseline). Further discussion can be found in Appendix G.3, which shows our method has very large positive effect sizes for Nodule and Synapse datasets.

Table 4: MedMNIST3D test accuracies. Mean over 3 runs; standard deviation in brackets. Parameter counts shown. Best result in each column is bold, second-best is underlined.

| Group | Model | Nodule $\uparrow$ | Synapse $\uparrow$ | Organ $\uparrow$ | #Params(M)$\downarrow$ |
|---|---|---|---|---|---|
| N/A | CNN | 0.873 (0.005) | 0.716 (0.008) | 0.920 (0.003) | 00.19 |
| Aug | CNN | 0.879 (0.007) | 0.761 (0.008) | 0.632 (0.005) | 00.19 |
| SO(3) | SCNN | 0.873 (0.002) | 0.738 (0.009) | 0.607 (0.006) | **00.13** |
| SO(3) | RPP | 0.801 (0.003) | 0.695 (0.037) | 0.936 (0.002) | 18.30 |
| SO(3) | PSCNN | 0.871 (0.001) | **0.770** (0.030) | 0.902 (0.006) | 04.17 |
| O(3) | SCNN | 0.868 (0.009) | 0.743 (0.004) | 0.902 (0.006) | 00.19 |
| O(3) | RPP | 0.810 (0.013) | 0.722 (0.023) | **0.940** (0.006) | 29.30 |
| O(3) | PSCNN | 0.873 (0.008) | 0.769 (0.005) | 0.905 (0.004) | 03.51 |
| $\mathrm{Sym}_{\mathrm{cube}}$ | Trivial rep | 0.867 (0.001) | 0.743 (0.002) | 0.571 (0.002) | 00.19 |
| $\mathrm{Sym}_{\mathrm{cube}}$ | Defining rep | 0.837 (0.013) | 0.756 (0.019) | 0.560 (0.025) | 00.19 |
| $\mathrm{Sym}_{\mathrm{cube}}$ | Ours (regular) | **0.887** (0.005) | **0.770** (0.002) | 0.642 (0.056) | 00.19 |

## 6.3 Autoregression Task, SMOKE, $G = C_4$

We evaluate our method on the SMOKE dataset, generated with PhiFlow (Holl et al., 2020) by Wang et al. (2022) (see Figure 7 for a visualisation). The task involves predicting future frames of a simulated smoke velocity field autoregressively. This task is only approximately equivariant to the symmetry group $C_4$ (90-degree rotations) due to the presence of non-equivariant buoyancy effects. Full details are provided in the appendix. Table 5(a) shows the test RMSE for each model on the metrics considered. All reported figures are imported from the original authors (Wang et al., 2022), except for ours, augmented CNN, and non-augmented CNN, for which we tune the learning rate. Our method outperforms all models except for PSCNN, which has slightly better scores, with more than

12 times the number of parameters. While our method uses the augmented training set, we note from comparing the two CNN baselines that this gives little advantage for this task. Further details can be found in Appendix G.4, showing very large positive effect sizes for all models except PSCNN.

## 6.4 AUTOENCODING TASK, 3D SHAPES, $G = O_h$

Finally, we test our method on the conformally transformed SHREC '11 dataset (Lian et al., 2011; Mitchel et al., 2022), following the pre-training and fine-tuning procedure of Mitchel et al. (2024). We apply our methodology with $O_h$ augmentations (octahedral symmetries) to pre-train a baseline autoencoder before fine-tuning the encoder for classification. As this is an autoencoding task, we symmetrize the equivariance loss to the decoder. NIso's kernel adds 18k parameters above our model, which has the same parameter count as the baseline autoencoder (AE). Results are given in Table 5(b). Our approach achieves 90.45% accuracy, outperforming the group-agnostic method NFT. Our method also surpasses NIso, a model capable of learning actions of infinite groups, even though our method uses only a finite subgroup. Further details can be found in Appendix G.5, with effect sizes showing equivalence between our method and NIso, and very large positive effect size for the other models.

Table 5: Mean over 3 runs; standard deviation in brackets. Parameter counts shown. Best result in each column is bold, second-best is underlined.

(a) Test RMSE for SMOKE dataset.

| Group | Model | Future $\downarrow$ | Domain $\downarrow$ | #Params(M) $\downarrow$ |
|-------|-------|------|--------|-----------|
| N/A | CNN | 0.81 (0.01) | 0.63 (0.00) | **0.25** |
| Aug | CNN | 0.83 (0.03) | 0.67 (0.06) | **0.25** |
| N/A | MLP | 1.38 (0.06) | 1.34 (0.03) | 8.33 |
| C4 | E2CNN | 1.05 (0.06) | 0.76 (0.02) | 0.62 |
| C4 | RPP | 0.96 (0.10) | 0.82 (0.01) | 4.36 |
| C4 | Lift | 0.82 (0.01) | 0.73 (0.02) | 3.32 |
| C4 | RGroup | 0.82 (0.01) | 0.73 (0.02) | 1.88 |
| C4 | RSteer | 0.80 (0.00) | 0.67 (0.01) | 5.60 |
| C4 | PSCNN | **0.77** (0.01) | **0.57** (0.00) | 3.12 |
| C4 | Ours | 0.78 (0.01) | 0.61 (0.01) | **0.25** |

(b) Test accuracy for SHREC '11 dataset.

| Model | Acc. $\uparrow$ |
|-------|------|
| NIso Mitchel et al. (2024) | 90.26 (1.27) |
| NFT Koyama et al. (2024) | 83.24 (2.03) |
| AE with aug | 69.36 (2.81) |
| MC Mitchel et al. (2022) | 86.5 |
| Ours | **90.45** (2.1) |

## 7 CONCLUSIONS

**Limitations and Future Work.** Our theoretical framework is developed for finite groups. However, we empirically demonstrate that our method can be applied effectively to tasks with continuous symmetries by selecting a rich finite subgroup; we employ this strategy to show that our model can outperform NIso, SCNN, RPP and PSCNN, which use continuous groups such as $SO(3)$, $O(3)$ and the conformal group, on the SHREC '11 and MedMNIST3D datasets. We expect this strategy could also be effectively employed to handle large finite groups (such as permutation groups), and in future work we aim to derive theoretical guarantees on the power of this approach. Our method requires data augmentation, although this is typically inexpensive when the group action on the input space is easy to construct, and our ablations with an augmented baseline show that our model delivers benefits far beyond augmentation. We would also like to explore how our model could enable augmentation directly in the latent space.

**Conclusions.** This work investigates an alternative path to building efficient equivariant models, focusing not on architectural design, but on the enforcement of a principled latent algebraic structure. We prove that for finite groups, this structure is the regular representation, which must appear almost surely in the latent space of any equivariant encoder. By enforcing this structure via a parameter-free auxiliary loss, our method achieves competitive or superior performance to SOTA models, while requiring in some cases significantly fewer parameters. Furthermore, we empirically show the optimality of the regular representation via ablations with the defining and trivial representations. Ultimately, our work suggests that for tasks with inherent (approximate) symmetry, directly enforcing the correct latent algebraic structure can be a more effective and efficient path to equivariance than designing complex, highly-parameterized architectures.

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

## A   CODE

The code to run all the experiments in this paper is available at the following location:

- https://anonymous.4open.science/r/parameter-free-approximate-equivariance-3352/

In the README file, we provide instructions to run the code and reproduce the results.

## B   NOTATION

Here we provide a comprehensive list of symbols and notational conventions used throughout the paper.

GENERAL MATHEMATICAL OBJECTS

$G$  A finite group.

$g, h$  Elements of the group $G$, e.g., $g \in G$.

$\mathbb{K}$  The base field, assumed to be either the real numbers $\mathbb{R}$ or the complex numbers $\mathbb{C}$.

$\mathcal{S}, \mathcal{A}$  General sets, denoted by calligraphic letters.

$V, W$  General vector spaces, denoted by uppercase Roman letters.

$v, w$  Elements (vectors) of a vector space, e.g., $v \in V$.

GROUP THEORY

$\alpha$  A group action on a set. The action of $g \in G$ on an element $s \in \mathcal{S}$ is written as $\alpha(g, s)$

$\rho_V$  A group representation on the vector space $V$, which is a linear group action on $V$.

$\rho_V(g)$  The invertible linear map associated with the group element $g \in G$. The action of $g$ on a vector $v \in V$ is written as $\rho_V(g)(v)$.

MACHINE LEARNING CONTEXT

$\mathcal{X}$  The input set.

$x$  A single input data point, $x \in \mathcal{X}$.

$\mathcal{Y}$  The output or label set.

$y$  A single output or label, $y \in \mathcal{Y}$.

$Z$  The latent space, viewed as a vector space (e.g., $Z = \mathbb{R}^d$).

$z$  A latent vector, $z \in Z$.

$E$  An encoder network.

$D$  A decoder network.

$\widehat{\rho}_Z$  A *learnable* representation on the latent space $Z$.

## C   GROUP ACTIONS AND REPRESENTATIONS

**Groups.**   A group $G$ is a set equipped with an associative and unital binary operation, such that every element has a unique inverse. Important families of groups include the following. The dihedral group $D_n$ is the group of symmetries of the regular polygon with $n$ sides, which we use in this paper for $n \geq 3$. The cyclic group $C_n$ is the groups of integers $\{0, \dots, n-1\}$ with addition modulo $n$. The permutation group $S_n$ is the group of permutations of an $n$-element set. We may define groups by presentations, which give generators and relations for the product; for example, the group $C_2$ can be defined by the presentation $\{1, a \mid a^2 = 1\}$. For any two groups $G, H$, we write $G \times H$ for the product group, whose elements are ordered pairs of elements of $G$ and $H$ respectively.

**Group representations.** A *representation* $\rho$ of a finite group $G$ on a vector space $V$ is a choice of linear maps $\rho(g) : V \to V$ for all elements $g \in G$, with the property that $\rho(e) = \mathrm{id}_V$ for the identity element $e \in G$, and such that $\rho(g)\rho(g') = \rho(gg')$ for all pairs of elements $g, g' \in G$. We define the *dimension* of $\rho$ to be $\dim(V)$, the dimension of the vector space $V$. There is a notion of equivalence of representations: given representations $\rho$ on $V$, and $\rho'$ on $V'$, they are *isomorphic* when there is an invertible linear map $L : V \to V'$ such that $L\rho(g) = \rho'(g)L$ for all $g \in G$. Given a subgroup $G \subseteq G'$, a representation of $G'$ yields a *restricted representation* on $G$ in an obvious way.

**Defining representations.** The concept of a defining representation is relevant for our ablation studies. While the term is context-dependent, it typically refers to a group's most natural or defining low-dimensional representation. For the permutation group $S_n$ this is the linearisation of its permutation action on the $n$-element set; that is, the $n$-dimensional representation given by its action on $\mathbb{K}^n$ by permuting the basis vectors. For the dihedral group $D_n$ ($n \geq 3$), the defining representation is the linearisation of its action on the $n$-element set of vertices. For the group $\mathrm{Sym}_{\mathrm{cube}}$ of orientation-preserving symmetries of the cube, the defining representation is the linearisation of its action on the 8-element set of vertices of the cube. We select these defining representations as a baseline as they provide a rich, geometrically intuitive alternative to the more abstract regular representation.

**Group actions.** A group may also have an *action* $\lambda$ on a set $\mathcal{S}$, a choice of functions $\lambda(g) : \mathcal{S} \to \mathcal{S}$ for all elements $g \in G$, such that $\lambda(e) = \mathrm{id}_{\mathcal{S}}$ and $\lambda(g)\lambda(g') = \lambda(gg')$. Such an action yields a representation of $G$ on $\mathbb{K}[\mathcal{S}]$ by linearisation, the *free $\mathbb{K}$-vector space* generated by $S$.

Some simple examples of representations include the *zero representation* on the zero-dimensional vector space, and the *trivial representation* $\rho_{\mathrm{triv}}$ on the 1-dimensional vector space $\mathbb{K}$, where $\rho_{\mathrm{triv}}(g) = \mathrm{id}_{\mathbb{K}}$ for all $g \in G$.

# D  INSIGHT INTO THE ALGEBRA LOSS

To give further insight into component (*iv*), suppose our goal is to learn a representation $\widehat{\rho}_Z$ of the group $C_2$, which has group presentation $\{1, a \mid a^2 = 1\}$. Then $\widehat{\rho}_Z$ should satisfy $\widehat{\rho}_Z(1) = \mathrm{id}$ and $\widehat{\rho}_Z(a^2) = (\rho_Z(a))^2 = \mathrm{id}$. To achieve this, we fix the parameter $\widehat{\rho}_Z(1) = \mathrm{id}$, and choose $\mathrm{ALG}_{C_2,d}$ and $\mathrm{REG}_{C_2,d}$ as follows, where $d = \dim(Z)$, the matrix $\mathrm{I}_d$ is the identity of size $d \times d$:

$$\mathrm{ALG}_{C_2,d} = \mathrm{MSE}\big(\widehat{\rho}_Z(a)^2,\ \mathrm{I}_d\big)$$
$$\mathrm{REG}_{C_2,d} = \mathrm{MSE}\big(\widehat{\rho}_Z(a),\ \widehat{\rho}_Z(a)^{-1}\big).$$

We note that when $\mathrm{ALG}_{C_2,d}$ equals zero then $\widehat{\rho}_Z(a)^2 = \mathrm{I}_d$, and hence $\mathrm{REG}_{C_2,d}$ will equal zero. In this sense, the regularisation term is algebraically redundant, but is found to improve training.

# E  PROOFS

We first introduce some basic definitions

**Definition 3.** Let $V$ be a vector space, and $W, W' \subseteq V$ be subspaces of $V$. $W$ and $W'$ are *linearly independent* if $W \cap W' = 0$.

**Definition 4.** Let $G$ act on a set $\mathcal{X}$ via the group action $\alpha_X$. The *orbit of $x \in \mathcal{X}$* is the set $\mathcal{O}_x = \{\alpha_{\mathcal{X}}(g)(x) \mid g \in G\}$. Given a vector space $Z$ and a function $E : \mathcal{X} \to Z$, we call the set $E(\mathcal{O}_x) = \{E(\alpha_{\mathcal{X}}(g)(x)) \mid g \in G\}$ the *embedded orbit of $x$ along $E$*. Two embedded orbits $E(\mathcal{O}_x), E(\mathcal{O}_{x'})$ are *linearly independent* if their spans are linearly independent, that is if $\mathrm{Span}(E_\theta(\mathcal{O}_x)) \cap \mathrm{Span}(E_\theta(\mathcal{O}_{x'})) = \{0\}$.

The proof of Lemma 7 adapts the argument in Nikolaou et al. (2025), which uses measure-theoretic properties of analytic functions to demonstrate that transformers are almost everywhere injective. Although our focus here is not on transformers, most of their results require only real analyticity, and thus can be easily adapted to our case. Intuitively, a measure $\mu$ on a set $X$ quantifies the 'size' or 'volume' of subsets within $X$ (see e.g., Fremlin (2000) for a foundational treatment). In the context of $\mathbb{R}^p$, the Lebesgue measure $\lambda$ corresponds to the standard notion of Euclidean volume (e.g., it assigns the unit hypercube a measure of 1).

**Notation.** If $f : X \to X$ is a function, we will write $f^{\circ T}$ to indicate the consecutive application of $f$ for $T$ times. If $\Theta$ is a set equipped with a measure $\mu$, we will write $\theta \sim \Theta$ to indicate that $\theta$ is a random draw of an element of $\Theta$ according to the measure $\mu$.

The following are well-known mathematical results and definitions.

**Proposition 5.** *Let $U \subseteq \mathbb{R}^m$ be open and connected, and let $f : U \to \mathbb{R}^n$ be an analytic function. If $f$ is not identically zero, then its zero set $Z(f) := \{x \in U \mid f(x) = 0\} = f^{-1}(0)$ has Lebesgue measure zero in $\mathbb{R}^m$, i.e. $\lambda(Z(f)) = 0$.*

**Definition 6.** Let $\mu$, $\nu$ be Borel measures on $\mathbb{R}^p$. We say that $\mu$ is *absolutely continuous* with respect to $\nu$, written $\mu \ll \nu$, if for every Borel set $U$ we have

$$\nu(U) = 0 \implies \mu(U) = 0 \tag{2}$$

Since the Lebesgue measure $\lambda$ is the standard notion of Euclidean volume, then we can intuitively understand that $\mu \ll \lambda$ just when the measure $\mu$ assigns zero measure to every set with zero Euclidean volume. In this way, measures $\mu$ with $\mu \ll \lambda$ may behave in ways that accord with our intuition. Any standard sampling measure $\mu$ used to generate initial parameters for a neural network (e.g. from the normal or uniform distributions) will be likely to satisfy this property.

The following critical lemma underlies our theoretical results, and we can explain it intuitively as follows. If we have some set of parameters $\mathcal{W} \subseteq \Theta$ for our neural network which is measure zero, then of course if we initialise the network with some parameters $\theta$ at random, the probability that $\theta \in \mathcal{W}$ is zero. But we then ask, if we update the parameters by gradient descent for finitely many steps, what is the probability that the optimised parameters are within the set $\mathcal{W}$?

**Lemma 7.** *Let $E_\theta$ be a parametrized function that is analytic for all parameters $\theta \in \Theta \subseteq \mathbb{R}^p$. Assume that the parameters are randomly initialized according to a distribution $\mu$ that is absolutely continuous with respect to the Lebesgue measure on $\mathbb{R}^p$, i.e. $\theta_0 \sim \mu$ with $\mu \ll \lambda$. Furthermore, assume an analytic loss function $\mathcal{L}$, and that the parameters are updated via gradient descent, i.e. $\theta_{t+1} := \Phi(\theta_t) := \theta_t - \eta \nabla \mathcal{L}(\theta_t)$, with $\eta \in (0, 1)$.*

*Let $\mathcal{W} \subseteq \Theta$ with $\lambda(\mathcal{W}) = 0$. Then, for all $T \in \mathbb{N}$, $\mu(\{\theta_0 \mid \Phi^{\circ T}(\theta_0) \in \mathcal{W}\}) = 0$.*

*Proof.* The proof uses standard analytic and measure-theoretic tools, and is an adaptation of the argument in the proof of (Nikolaou et al., 2025, Theorem C.1). Let $\mathcal{W} \subseteq \Theta$ with $\lambda(\mathcal{W}) = 0$. First, we apply (Nikolaou et al., 2025, Lemma C.6), that $\lambda(\Phi^{-1}(\mathcal{W})) = 0$. (Note that in this paper, Lemma C.6 assumes the context of a transformer; however the proof only uses analyticity of the components, and thus the result holds in our case). Then, because $\mu \ll \lambda$, it follows that $\mu(\{\theta_0 \mid \Phi(\theta_0) \in \mathcal{W}\}) = \mu(\Phi^{-1}(W)) = 0$. By applying the same argument $T$ times, we find $\mu(\{\theta_0 \mid \Phi^{\circ T}(\theta_0) \in \mathcal{W}\}) = 0$. $\square$

**Theorem 1.** Let $G$ be a finite group acting on a set $\mathcal{A}$ with action $\alpha_\mathcal{A}$, and on a vector space $Z$ with a representation $\rho_Z$, with $\dim(Z) \geq |G|$. Suppose that the group acts freely and transitively on some subset $\mathcal{S} \subseteq \mathcal{A}$. If $E : \mathcal{A} \to Z$ is an equivariant function and $E(\mathcal{S})$ is full rank, then $Z$ contains the regular representation almost surely.

*Proof.* Let $\mathbb{R}[\mathcal{S}]$ denote the vector space of all formal linear combinations of $\mathcal{S}$ with coefficients in $\mathbb{R}$. Because $\alpha_\mathcal{A}|_\mathcal{S}$ is free and transitive it must be equivalent to the action of $G$ on itself, and hence its linearization $\mathbb{R}[\mathcal{S}]$ carries the structure of the regular representation. We write this representation explicitly as $\rho_{\mathbb{R}[\mathcal{S}]}(g)(\sum_s a_s s) := \sum_s a_s \alpha_\mathcal{A}(g)(s)$. Now, we can define the linear map $\tilde{E}^\mathcal{S} : \mathbb{R}[\mathcal{S}] \to Z$ by $(\sum_s a_s s) \mapsto \sum_s a_s E(s)$. Because $E : \mathcal{A} \to Z$ is equivariant, we conclude that $\tilde{E}^\mathcal{S} : \mathbb{R}[\mathcal{S}] \to Z$ is equivariant. Denote $V := \mathrm{Im}(\tilde{E}^\mathcal{S}) = \mathrm{Span}_\mathbb{R}\{E(s) \mid s \in \mathcal{S}\}$, which is a linear subspace of $Z$ of dimension at most $|G|$. By the first isomorphism theorem for representations (Fulton & Harris, 2004), we have $V \cong \mathbb{R}[\mathcal{S}]/\mathrm{Ker}(\tilde{E}^\mathcal{S})$. Finally, by assumption we have that $E(\mathcal{S})$ is full rank, implying that $\mathrm{Ker}(\tilde{E}^\mathcal{S})$ is trivial and that $V$ is isomorphic to the regular representation $\mathbb{R}[\mathcal{S}]$. Furthermore, we note that when $\dim Z = |G|$, it follows that $Z$ must be isomorphic to the regular representation itself. $\square$

We now combine Lemma 7 and Theorem 1 to obtain guarantees on the existence of regular representations in the latent space.

**Theorem 2.** Let $G$ be a finite group, $\mathcal{X} \subseteq \mathbb{R}^n$ an open and connected set, $Z$ a vector space with $\dim(Z) \geq |G|$, and $E_\theta : \mathcal{X} \to Z$ a function which is analytic on its domain $\mathcal{X}$ for all parameter values $\theta \in \Theta \subseteq \mathbb{R}^p$. Let $G$ act on a set $\mathcal{A} \subseteq \mathcal{X}$ with action $\alpha_\mathcal{A}$, and on $Z$ with a representation $\rho_Z$. Suppose that $\alpha_A$ is free and transitive on some $\mathcal{S} \subseteq \mathcal{A}$. Furthermore, suppose that the parameters are randomly initialized and updated by gradient descent with respect to an analytic loss function $\mathcal{L}$:

$$\theta_0 \sim \mu, \mu \ll \lambda \text{ the Lebesgue measure on } \mathbb{R}^p$$
$$\theta_{t+1} := \Phi(\theta_t) := \theta_t - \eta\nabla\mathcal{L}(\theta_t) \text{ with } \eta \in (0,1)$$

yielding an equivariant analytic function. Then, either $Z$ contains the regular representation almost surely, or $E_\theta(\mathcal{S})$ is rank deficient for all possible parameterizations $\theta \in \Theta$.

*Proof.* Adopting the setup from the proof of Theorem 1, we must now show that $Z$ contains the regular representation almost surely, i.e. that $\mathrm{Ker}(\tilde{E}_\theta^\mathcal{S})$ is trivial almost surely or constantly zero for all $\theta \in \Theta$ and $\mathcal{S} \subseteq \mathcal{A}$ on which $\alpha_\mathcal{A}$ acts transitively. The linear function $\tilde{E}_\theta^\mathcal{S}$ is fully specified by its action on the basis $\mathcal{S}$, i.e. $\{E_\theta(s) \,|\, s \in \mathcal{S}\}$. Because $V$ has dimension at most $G$, we may embed each of the $E_\theta(s)$ into $y_s \in \mathbb{R}^{|G|}$. Therefore, in matrix representation, $\tilde{E}_\theta^\mathcal{S}$ is obtained by collecting the vectors $\{y_s \,|\, s \in \mathcal{S}\}$ in a $|G| \times |G|$ matrix $M_\theta^\mathcal{S}$. The condition for the kernel to be trivial is $\det M_\theta^\mathcal{S} \neq 0$. Now, $\det M_\theta^\mathcal{S}$ is an analytic function, as each entry of $M_\theta^\mathcal{S}$ is analytic by analyticity of $E_\theta^\mathcal{S}$, and the determinant is a polynomial and hence analytic. Therefore, we get that the set $\mathcal{W} := \{\theta \in \Theta \,|\, \det M_\theta^\mathcal{S} = 0\}$ has measure zero with respect to $\mu$ by Proposition 5 and absolute continuity of $\mu \ll \lambda$, or it is constantly zero for all $\theta \in \Theta$. By Lemma 7, we then get that $\mu(\{\theta_0 \in \Theta \,|\, \Phi^{\circ T}(\theta_0) \in \mathcal{W}\}) = 0$, meaning that $\det M \neq 0$ with probability 1. $\qquad\square$

## E.1 THE ANALYTICITY CONDITION

We remark that, as observed by Nikolaou et al. (2025), most standard modules used in neural network, such as linear layers, layer norm, skip connections, convolutions, attention, and others are analytic. The same holds for many commonly used activation functions, such as tanh, sigmoid, softplus, softmax, SiLU, GELU, SwiGLU. Therefore, the analytic condition does not heavily restrict our analysis. For example, Nikolaou et al. (2025) highlight that decoder-only transformers are analytic. However, others activation functions are only piece-wise analytic, e.g. ReLU, LeakyReLU, ELU. For this reason, we repeat the TMNIST and MNIST experiments from Section 4.2 with non-analytic encoders to empirically test whether our conclusions hold for this class of networks. We find this to be the case, and we discuss it in Appendix F.3.

# F EXPLORATORY EXPERIMENTS

This section is organized as follows:

- Section F.1 describes how we extract the embedded orbits and check their linear independence to get the number of linearly independent orbits.
- Section F.2 contains further details for the exploratory experiments, including hyperparameters and regularization terms for the algebra loss.
- Section F.3 repeats the TMNIST and MNIST experiments from Section 4.2 but for non-analytic encoders.
- Section F.4 repeats the TMNIST experiment from Section 4.2 by varying the depth of the layer considered as $Z$.
- Section F.5 repeats the TMNIST experiment from Section 4.2 by changing the initialization scheme for the learnable group action $\widehat{\rho}_Z$.

## F.1 EXTRACTING EMBEDDED ORBITS AND CHECKING THEIR LINEAR INDEPENDENCE

We describe how we extract the embedded orbits and how we compute their linear independence. Let $E : \mathcal{X} \to Z$ denote the encoder, and $G = \{g_1, \ldots, g_n\}$ the finite group considered. First, we compute the embedded orbit as $E(\mathcal{O}_x) = \{\widehat{\rho}_Z(g_i)(E(x))\}_{i\in\{1,\ldots,n\}} \subseteq Z$ with $|E(\mathcal{O}_x)| = |G|$.

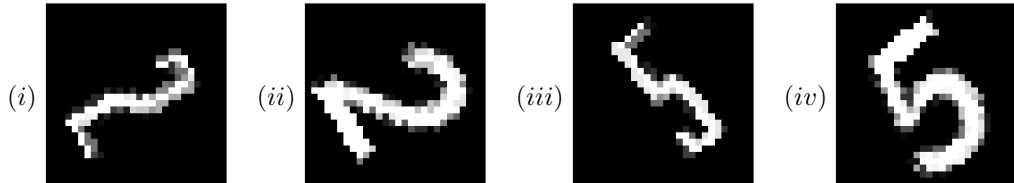

Figure 5: Examples of our augmented training dataset for the TMNIST experiment, from the chosen fonts 'Bahianita-Regular' $(i)$, $(iii)$ and 'IBMPlexSans-MediumItalic' $(ii)$, $(iv)$.

Then, given embedded orbits $E(\mathcal{O}_{x_1}), \ldots, E(\mathcal{O}_{x_m})$, we collect all vectors in their union in a matrix $K \in \mathbb{R}^{m|G| \times d}$. These orbits are linearly independent if the matrix $K$ is full rank, which is computed by checking that all its singular values are non-zero.

Each number in the 'Orbits' columns in the Tables from Sections 4.2 and Appendix F.3, F.4 and F.5 is the maximum number of linearly independent orbits found by randomly sampling combinations of training samples $x \in \mathcal{X}$. For each run, we sample 500 different combinations.

### F.2 FURTHER DETAILS FOR THE EXPLORATORY EXPERIMENTS

Here we give details of the exploratory experiments we describe in Section 4. These use the TMNIST, MNIST and CIFAR10 datasets to determine the optimal representation on the latent space. Sections F.2.1, F.2.2 and F.2.3 provide details of the architectures and regularisation terms used for each of these experiments. In all runs, we use the Adam optimiser Kingma & Ba (2017) with default parameters $(\beta_1, \beta_2) = (0.9, 0.999)$, and report additional hyperparameters in Table 6. These were chosen through a manual tuning process.

### F.2.1 TMNIST AUTOENCODER, $G = C_2$

This experiment uses the TMNIST dataset Magre & Brown (2022) of digits rendered in a variety of typefaces. We select a data subset corresponding to just two typefaces 'IBMPlexSans-MediumItalic' and 'Bahianita-Regular', and augment with 180° rotations. We give some examples of our augmented dataset in Figure 5. The group we use here is $C_2 = \{1, a \mid a^2 = 1\}$ and, for a data point $x$, we define the group action $\rho_{\mathcal{X}}(a)(x)$ to be the data point with the font swapped, but the rotation and scaling unchanged. In particular, with reference to images Figure 5(i)–(iv), we have $\rho_{\mathcal{X}}(a)(i) = (ii)$, $\rho_{\mathcal{X}}(a)(ii) = (i)$, $\rho_{\mathcal{X}}(a)(iii) = (iv)$ and $\rho_{\mathcal{X}}(a)(iv) = (iii)$. For this experiment we set $L_{\text{task}} = $ MSE, and we use a simple CNN autoencoder with hyperparameters given in Table 6. The architectural details can be found on the provided repository.

Table 6: Hyperparameters for exploratory experiments.

| Experiment | Latent dim. | $\lambda_a$ | $\lambda_t$ | $\lambda_e$ | LR | Batch Size |
|---|---|---|---|---|---|---|
| TMNIST $C_2$ | 8 | 1.0 | 0.5 | 1 | 0.003 | 64 |
| MNIST $D_3$ | 18 | 0.5 | 0.495 | 0.005 | 0.003 | 64 |
| CIFAR10 $C_4$ | 16 | 1.0 | 25 | 0.25 | 0.003 | 64 |

We use the following regularization term:

$$\text{REG}_{C_2, d} = \text{MSE}(\widehat{\rho}_Z(a), \widehat{\rho}_Z(a)^{-1}) \tag{3}$$

Here $\widehat{\rho}_Z(a)^{-1}$ is computed with $\widehat{\rho}_Z(a)^{-1} = \texttt{torch.linalg.solve}(\widehat{\rho}_Z(a), \text{I}_d)$ for efficiency and numerical stability. We found empirically that this regularization helps to stabilize the training of $\widehat{\rho}_Z(a)$, allowing us to achieve lower values for the algebra loss.

### F.2.2 MNIST AUTOENCODER, $G = D_3$

This experiment uses the MNIST dataset Deng (2012) of handwritten digits. The group considered is $D_3 = \{e, r, r^2, r^3, s, rs \mid r^3 = e, s^2 = e, rsrs = e\}$, and on the input space we define the group action such that $\rho_{\mathcal{X}}(r)(x)$ is the counterclockwise rotation of $x$ by 120 degrees, and $\rho_{\mathcal{X}}(s)(x)$ is the image generated by flipping $x$ about the vertical axis. For this experiment, we set $L_{\text{task}} = \text{MSE}$, and use a simple MLP autoencoder with hyperparameters given in Table 6. The architectural details can be found on the provided repository.

We use the following regularization term:

$$\text{REG}_{D_3,d} = -0.995\,\text{MSE}(\widehat{\rho}_Z(r)\widehat{\rho}_Z(s)\widehat{\rho}_Z(r)\widehat{\rho}_Z(s), \text{I}_d) \tag{4}$$

We determined empirically that this regularization dampens the interaction between the matrices $\widehat{\rho}_Z(r)$ and $\widehat{\rho}_Z(s)$ in a way that improves training. Low final values of the algebra loss reported in Table 1 give evidence that we still obtain a high-quality representation despite this damping.

### F.2.3 CIFAR10 CLASSIFIER, $G = C_4$

This experiment uses the CIFAR10 dataset Krizhevsky (2009) of 32x32 images organised in 10 classes: airplane, automobile, bird, cat, deer, dog, frog, horse, ship, truck. The group considered is the cyclic group of size four $C_4$ of addition on the set $\{0, 1, 2, 3\}$ modulo 4. The element 1 is a generator for this group, and for an input vector $x$, we define the group action such that $\rho_{\mathcal{X}}(1)(x)$ is the rotation of $x$ by 90 degrees counterclockwise. For this experiment we set $L_{\text{task}} = \text{CrossEntropy}$, and use a simple CNN classifier with hyperparameters given in Table 6. The architectural details can be found on the provided repository.

The regularization term used is the following:

$$\text{REG}_{C_4,d} = \text{MSE}(\widehat{\rho}_Z(1)^3, \widehat{\rho}_Z(1)^{-1}) \tag{5}$$

Here, $\widehat{\rho}_Z(1)^{-1}$ is computed with $\widehat{\rho}_Z(1)^{-1} = \texttt{torch.linalg.solve}(\widehat{\rho}_Z(1), \text{I}_d)$ for efficiency and numerical stability. We determined empirically that this regularization helps to stabilize the training of $\widehat{\rho}_Z(1)$ and the behavior of its inverse.

### F.3 EXPLORATORY EXPERIMENTS FOR NON-ANALYTIC ENCODERS

In this section, we repeat the same experiments for TMNIST $C_2$ and MNIST $D_3$ from Section 4.2 but for non-analytic encoders. In particular, we use the same architecture but we replace the $\texttt{tanh}$ activation function with ReLU. Table 7 shows similar results as the fully analytic encoders (Table 1), suggesting empirically that the optimization process avoids any potentially degenerate regions.

Table 7: Piece-wise analytic encoder experiments. Left, TMNIST autoencoder task, learned representations of $C_2$ on latent space. Right, MNIST autoencoder task, learned representations of $D_3$ on latent space.

| | Irrep. counts | | | | | | Irrep. counts | | | | | |
|-----|-----|-----|-----------|-----------|------|-----|------|------|------|-----------|-----------|------|
| Run | $-1$ | $+1$ | Alg. loss | Eq. loss | Orbs. | Run | Triv | Sgn | Std | Alg. loss | Eq. loss | Orbs. |
| 1 | 4 | 4 | $5.7\times10^{-5}$ | $4.6\times10^{-3}$ | 4 | 1 | 3.01 | 3.01 | 5.99 | $1.2\times10^{-3}$ | $1.3\times10^{-2}$ | 3 |
| 2 | 3 | 5 | $6.7\times10^{-9}$ | $6.6\times10^{-6}$ | 3 | 2 | 2.98 | 2.98 | 6.01 | $6.1\times10^{-4}$ | $2.3\times10^{-2}$ | 3 |
| 3 | 4 | 4 | $2.7\times10^{-8}$ | $2.5\times10^{-5}$ | 4 | 3 | 3.32 | 3.36 | 5.66 | $3.1\times10^{-2}$ | $1.4\times10^{-2}$ | 3 |
| 4 | 4 | 4 | $2.3\times10^{-9}$ | $4.2\times10^{-6}$ | 4 | 4 | 3.03 | 3.31 | 5.69 | $1.4\times10^{-2}$ | $1.2\times10^{-2}$ | 3 |
| 5 | 3 | 5 | $6.0\times10^{-9}$ | $1.9\times10^{-5}$ | 3 | 5 | 2.98 | 2.98 | 6.02 | $8.5\times10^{-4}$ | $1.3\times10^{-2}$ | 3 |

### F.4 EXPLORATORY EXPERIMENTS AT DIFFERENT LAYER DEPTHS

In this section, we repeat the TMNIST experiment from Section 4.2 for different layer depths. In the original experiment, we choose to study equivariance with respect to the layer $Z$ chosen as the central hidden layer (the output layer of the encoder). Table 8 shows the results of choosing $Z$ as the first or last hidden layer. The results are similar to those in Section 4.2: each linearly independent embedded orbit corresponds to a copy of the regular representation, and the network tends to learn a multiple of it.

Table 8: TMNIST experiment with $Z$ at different depths. Left: $Z$ is taken as the first hidden layer; Right: $Z$ is taken as the final hidden layer.

| | Irrep. counts | | | | | | Irrep. counts | | | | |
|-----|-----|-----|----------|---------|------|-----|-----|-----|----------|---------|------|
| Run | $-1$ | $+1$ | Alg. loss | Eq. loss | Orbs. | Run | $-1$ | $+1$ | Alg. loss | Eq. loss | Orbs. |
| 1 | 3 | 5 | $4.9\times10^{-10}$ | $1.1\times10^{-4}$ | 3 | 1 | 3 | 5 | $6.8\times10^{-9}$ | $4.5\times10^{-4}$ | 3 |
| 2 | 3 | 5 | $4.2\times10^{-9}$ | $1.6\times10^{-4}$ | 3 | 2 | 4 | 4 | $9.3\times10^{-10}$ | $3.7\times10^{-4}$ | 4 |
| 3 | 4 | 4 | $1.0\times10^{-10}$ | $6.2\times10^{-5}$ | 4 | 3 | 3 | 5 | $1.8\times10^{-8}$ | $4.5\times10^{-4}$ | 3 |
| 4 | 4 | 4 | $2.3\times10^{-6}$ | $3.0\times10^{-5}$ | 4 | 4 | 4 | 4 | $6.9\times10^{-10}$ | $4.6\times10^{-4}$ | 4 |
| 5 | 3 | 5 | $9.0\times10^{-10}$ | $1.2\times10^{-4}$ | 3 | 5 | 4 | 4 | $2.3\times10^{-8}$ | $4.4\times10^{-4}$ | 4 |

### F.5 Exploratory experiment with different initialization

In this section, we repeat the TMNIST experiment from Section 4.2 but with a different initialization scheme. While Table 1 shows results for $\widehat{\rho}_Z$ initialized according to a normal distribution $\mathcal{N}(\mathbf{0}, \mathbf{I}_d)$, Table 9 shows results for the same experiment with $\widehat{\rho}_Z$ initialized close to the identity as $\mathbf{I}_d + \mathcal{N}(\mathbf{0}, \mathbf{I}_d)$.

The results confirm Theorem 1, as each linearly independent embedded orbit contributes one copy of the regular representation. However, the network typically does not learn a representation that consists entirely of a multiple of the regular representation. We observe that the trivial representation, corresponding to the eigenvalue $+1$ of $\widehat{\rho}_Z$ is over-represented. We hypothesize that the strong priming given by the initialization prevents a full exploration of the parameter space. To establish the practical advantage of the regular representation, we provide ablations with the trivial representation in controlled settings (Sections 6.1 and 6.2).

Table 9: TMNIST experiment with $\widehat{\rho}_Z$ initialized close to the identity.

| | Irrep. counts | | | | |
|-----|-----|-----|----------|---------|------|
| Run | $-1$ | $+1$ | Alg. loss | Eq. loss | Orbs. |
| 1 | 2 | 6 | $3.1\times10^{-5}$ | $2.9\times10^{-4}$ | 2 |
| 2 | 2 | 6 | $9.5\times10^{-4}$ | $0.3\times10^{-4}$ | 2 |
| 3 | 2 | 6 | $9.5\times10^{-4}$ | $1.6\times10^{-4}$ | 2 |
| 4 | 2 | 6 | $4.6\times10^{-5}$ | $1.8\times10^{-4}$ | 2 |
| 5 | 2 | 6 | $2.4\times10^{-5}$ | $9.1\times10^{-5}$ | 2 |

## G MAIN EXPERIMENTS

Here we give details of the main experiments we describe in Section 6, which test our model of Section 5 on tasks using the DDMNIST, MedMNIST, SMOKE and SHREC'11 datasets. Section G.1 discusses Cohen's $d$-statistic, which we use to assess the effect size of our intervention. Sections G.2, G.3, G.4 and G.5 provide details of the datasets, architectures and hyperparameters that we use, together with an effect size analysis. In all runs we use the Adam optimizer Kingma & Ba (2017) with default parameters $(\beta_1, \beta_2) = (0.9, 0.999)$, with weight decay set to 0 for DDMNIST and MedMNIST, and set to $4 \times 10^{-4}$ for SMOKE.

### G.1 COHEN'S $d$-STATISTIC

Cohen's $d$-statistic is a widely-adopted metric (Miranda et al., 2025; Huang et al., 2024; Gundersen et al., 2023; Karandikar et al., 2021; Hermann et al., 2024) to assess effect size, i.e. the meaningfulness of the difference between distributions. In particular, Cohen's $d$ quantifies the difference between two distributions in standard deviation units. Commonly used thresholds in machine learning are the following (Hermann et al., 2024):

- $|d| < 0.5$, small effect
- $0.5 \leq |d| < 0.8$, medium effect

- $0.8 \leq |d| < 1.2$, large effect

- $1.2 \leq |d|$, very large effect

Suppose we are given $n_1$ and $n_2$ observations of two distributions, with means $\overline{x}_1$ and $\overline{x}_2$, and standard deviations $s_1$ and $s_2$ respectively. Cohen's $d$ is then defined as follows:

$$d = \frac{\overline{x}_1 - \overline{x}_2}{s}, \qquad s = \sqrt{\frac{(n_1 - 1)s_1^2 + (n_2 - 1)s_2^2}{n_1 + n_2 - 2}} \tag{6}$$

To assess the effect size of our model, we choose $\overline{x}_1$ to be the mean result of our model on a particular task, and $\overline{x}_2$ to be the mean result of a benchmark model. When reported in the tables below, we choose the sign of the effect value so that a positive value indicates our model performed better.

### G.2 DDMNIST EXPERIMENTS

**Data preparation.** We follow closely the setup of the originators Veefkind and Cesa Veefkind & Cesa (2024). To generate this dataset, pairs of MNIST 28x28 images are chosen uniformly at random, and independently augmented according to the corresponding group action for $G \in \{C_4, C_2, D_4\}$ as per Table 10. We give an example in Figure 6. To ensure comparability of our results with the original paper, for $G \in \{C_4, D_4\}$ we follow their method of introducing interpolation artefacts by rotating each digit image by a random angle $\theta \in [0, 2\pi)$, and then rotating it back by $-\theta$; for $G = C_2$ these interpolation artefacts are not added, in line with the original paper. Finally, the two images are concatenated horizontally, and padded so that the final image is $56 \times 56$. In this way, we obtain a dataset of 10,000 images with labels in the set $\{(0, 0), (0, 1), \ldots, (9, 9)\}$.

Table 10: Symmetry groups and their actions on DDMNIST.

| Group | Type | Generators | Size |
|-------|------|------------|------|
| $C_4$ | Cyclic | 90° rotation | 4 |
| $C_2$ | Dihedral | Horizontal reflection | 2 |
| $D_4$ | Dihedral | Horizontal reflection and 90° rotation | 8 |

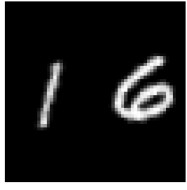 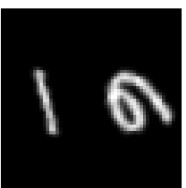

Before augmentation    After augmentation

Figure 6: Examples of training data for the DDMNIST experiment with $G = D_4$. The left figure shows concatenated MNIST digits, and the right figure shows the result after a random augmentation. In this instance, the left digit is augmented with a reflection about the vertical axis, and the right digit is augmented with a clockwise 90-degree rotation.

**Architecture.** We use the same CNN architecture as in Veefkind and Cesa Veefkind & Cesa (2024), except that the final convolutional layer has an increased number of filters, from 48 to 66. We make this change so that we can fit a copy of the regular representation of $D_4 \times D_4$. To ensure a fair comparison, the results reported in Table 3, including those for SCNN, RPP, etc, are those obtained with the increased number of filters, which we found marginally improved performance. Furthermore, we use a different learning rate for the CNN model, as we found that this increased performance and ensured a more meaningful baseline comparison. The CNN architectural details can be found on the provided repository.

**Hyperparameters.** We report the hyperparameters used for the CNN and our model for the DDM-NIST experiments in Table 11. These hyperparameters were chosen after a grid search with the following values: learning rate $\in \{0.001, 0.005, 0.0001, 0.0005, 0.00001, 0.00005\}$, and equivariance coupling strength $\lambda \in \{0.5, 1, 1.5, 2\}$. All other hyperparameters match those used by Veefkind and Cesa.

Table 11: Hyperparameters for DDMNIST experiments.

|  | $C_4$ | | $C_2$ | | $D_4$ | |
|---|---|---|---|---|---|---|
|  | LR | $\lambda$ | LR | $\lambda$ | LR | $\lambda$ |
| CNN | 0.0005 | - | 0.001 | - | 0.0005 | - |
| Standard rep | - | - | - | - | 0.0005 | 1 |
| Ours (regular) | 0.001 | 2 | 0.001 | 1 | 0.0005 | 1 |

**Effect size analysis.** We report the effect size of our intervention in Table 12. For each model, the 'Effect' column reports the Cohen $d$-value, comparing that model against 'Ours' with the regular representation. We observe that, for each model considered, there is at least one task where the difference with our model is very large according to Cohen's $d$ statistic (Appendix G.1).

Table 12: DDMNIST test accuracies and effect sizes. Mean over 3 runs; standard deviation in brackets. Best result in each column is bold, second-best is underlined. For $C_2, C_4$ the defining representation is equivalent to the regular representation and so is omitted. Effect values compare to 'Ours (regular)', and a positive value means ours performed better. The annotations *, **, *** indicate medium, large and very large effect sizes respectively.

| Model | $C_4 \uparrow$ | Effect | $C_2 \uparrow$ | Effect | $D_4 \uparrow$ | Effect |
|---|---|---|---|---|---|---|
| CNN | 0.907 (0.004) | 2.0*** | 0.938 (0.006) | 1.8*** | 0.800 (0.001) | 43.0*** |
| SCNN | 0.484 (0.008) | 68.2*** | 0.474 (0.003) | 133.8*** | 0.431 (0.010) | 60.6*** |
| Restriction | 0.914 (0.007) | 0.2 | 0.890 (0.007) | 10.0*** | 0.837 (0.020) | 2.2*** |
| RPP | 0.908 (0.022) | 0.4 | 0.903 (0.009) | 6.3*** | 0.827 (0.020) | 2.9*** |
| PSCNN | 0.909 (0.007) | 1.1** | 0.871 (0.016) | 6.5*** | 0.842 (0.011) | 3.3*** |
| Trivial rep | 0.874 (0.004) | 10.0*** | 0.938 (0.007) | 1.6*** | 0.819 (0.004) | 15.5*** |
| Defining rep | – | | – | | 0.838 (0.010) | 4.2*** |
| Ours (regular) | **0.915** (0.004) | | **0.947** (0.004) | | **0.868** (0.002) | |

### G.3 MEDMNIST EXPERIMENTS

**Data preparation.** For this experiment, we use three subsets of the MedMNIST dataset Yang et al. (2023), in line with Veefkind and Cesa Veefkind & Cesa (2024): Nodule3D, Synapse3D and Organ3D, each containing 3D images of size 28x28x28. Nodule3D is a public lung nodule dataset, containing 3D images from thoracic CT scans; for this dataset, the task is to classify each nodule as benign or malignant. Synapse3D contains 3D images obtained from an adult rat with a multi-beam scanning electron microscope; the task is to classify whether a synapse is excitatory or inhibitory. Organ3D is a classification task for a 3D images of human body organs, with the following labels: liver, right kidney, left kidney, right femur, left femur, bladder, heart, right lung, left lung, spleen and pancreas.

For augmentations, we choose the octahedral group of orientation-preserving rotational symmetries of the cube, which is isomorphic to the permutation group $S_4$. We define its action $\rho_{\mathcal{X}}(g)$ on a 3D image $x$ by applying the corresponding rotational symmetry of the cube. Specifically, we parameterise $g$ as a tuple $(l, \theta)$ where $l = (x, y, z)$ specifies a rotation axis and $\theta$ specifies the rotation angle about the axis $l$ to obtain 24 rotation matrices each with size $3 \times 3$, one for each of the 24 elements of $S_4$. In summary, we have rotation matrices corresponding to the following tuples:

Identity (1) $(l, 0)$ for any $l$.
Coord-axis (9) $(l, \theta)$ for $l \in \{(1, 0, 0), (0, 1, 0), (0, 0, 1)\}$ and $\theta \in \{\frac{\pi}{2}, \pi, \frac{3\pi}{2}\}$.

Edge-mid (6)  $(l, \theta)$ for $l \in \{(1, 1, 0), (1, -1, 0), (1, 0, 1), (1, 0, -1), (0, 1, 1), (0, 1, -1)\}$
and $\theta = \pi$.

Body-diag (8)  $(l, \theta)$ for $l \in \{(1, 1, 1), (1, 1, -1),$
$(1, -1, 1), (-1, 1, 1)\}$ and $\theta \in \{\frac{2\pi}{3}, \frac{4\pi}{3}\}$.

**Architecture.**    For these experiments we use the same CNN-based ResNet architecture as Veefkind and Cesa Veefkind & Cesa (2024). This is formed from seven 3D convolutional layers, formed into 3 blocks with residual connections, along with batch normalisation and pooling. The architectural details can be found on the provided repository.

**Hyperparameters.**    We report the hyperparameters used for the baseline with $S_4$ augmentations, and for our model in the MedMNIST experiments in Table 13.    These hyperparameters were chosen after a grid search with the following values:  learning rate $\in$ $\{0.001, 0.005, 0.0001, 0.0005, 0.00001, 0.00005\}$,  and equivariance coupling strength $\lambda$ $\in$ $\{0.5, 1, 1.5, 2\}$. All other hyperparameters are the same as those used by Veefkind and Cesa.

Table 13: Hyperparameters for MedMNIST experiments.

|  | Nodule3D | | Synapse3D | | Organ3D | |
| --- | --- | --- | --- | --- | --- | --- |
|  | LR | $\lambda$ | LR | $\lambda$ | LR | $\lambda$ |
| CNN (Augmented) | 0.00005 | - | 0.0001 | - | 0.0001 | - |
| Ours | 0.00005 | 1 | 0.0001 | 1 | 0.0001 | 2 |

**Effect size analysis.**    We report the effect size of our intervention in Table 14. For each model, the 'Effect' column reports the Cohen $d$-value comparing that model against 'Ours' with the regular representation.    We observe that, for each model considered, there is at least one task where the difference with our model is very large according to Cohen's $d$ statistic (Appendix G.1).

Table 14: MedMNIST3D test accuracies and effect sizes. Mean over 3 runs; standard deviation in brackets. Parameter counts shown. Best result in each column is bold, second-best is underlined. Effect values compare to 'Ours (regular)', and a positive value means ours performed better. The annotations *, **, *** indicate medium, large and very large effect sizes respectively.

| Group | Model | Nodule ↑ | Effect | Synapse ↑ | Effect | Organ ↑ | Effect |
| --- | --- | --- | --- | --- | --- | --- | --- |
| N/A | CNN | 0.873 (0.005) | 2.80*** | 0.716 (0.008) | 9.26*** | 0.920 (0.003) | -7.01*** |
| Aug | CNN | 0.879 (0.007) | 1.32*** | 0.761 (0.008) | 1.54*** | 0.632 (0.005) | 0.25 |
| SO(3) | SCNN | 0.873 (0.002) | 3.68*** | 0.738 (0.009) | 4.91*** | 0.607 (0.006) | 0.88** |
| SO(3) | RPP | 0.801 (0.003) | 20.86*** | 0.695 (0.037) | 2.86*** | 0.936 (0.002) | -7.42*** |
| SO(3) | PSCNN | 0.871 (0.001) | 4.44*** | **0.770** (0.030) | 0.00 | 0.902 (0.006) | -6.53*** |
| O(3) | SCNN | 0.868 (0.009) | 2.61*** | 0.743 (0.004) | 8.54*** | 0.902 (0.006) | -6.53*** |
| O(3) | RPP | 0.810 (0.013) | 7.82*** | 0.722 (0.023) | 2.94*** | **0.940** (0.006) | -7.48*** |
| O(3) | PSCNN | 0.873 (0.008) | 2.10*** | 0.769 (0.005) | 0.26 | 0.905 (0.004) | -6.62*** |
| $Sym_{cube}$ | Trivial rep | 0.867 (0.001) | 5.55*** | 0.743 (0.002) | 13.50*** | 0.571 (0.002) | 1.79*** |
| $Sym_{cube}$ | Defining rep | 0.837 (0.013) | 5.08*** | 0.756 (0.019) | 1.04** | 0.560 (0.025) | 1.89*** |
| $Sym_{cube}$ | Ours (regular) | **0.887** (0.005) |  | **0.770** (0.002) |  | 0.642 (0.056) |  |

## G.4   SMOKE EXPERIMENT

**Data preparation.**    Here we use the SMOKE dataset of Wang et al. Wang et al. (2022), which consists of smoke simulations with varying intial conditions and external forces presented as grids of $(x, y)$ components of a velocity field (see Figure 7 for a visualisation). The task is to predict the next 6 frames of the simulation given the first 10 frames only. We evaluate each model on two metrics: Future, where the test set contains future extensions of instances in the training set; and Domain, where the test and training sets are from different instances. In line with Wang et al. (2022), we consider the group $C_4$ acting on the data by 90° rotations and reorientation of the velocity field, as illustrated in Figure 8.

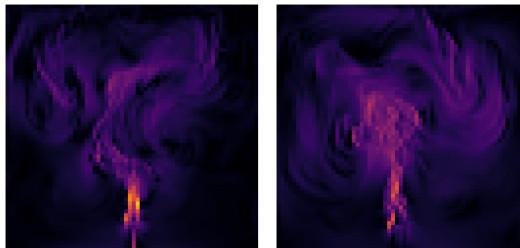

Figure 7: Approximately equivariant dynamics of smoke plumes Holl et al. (2020).

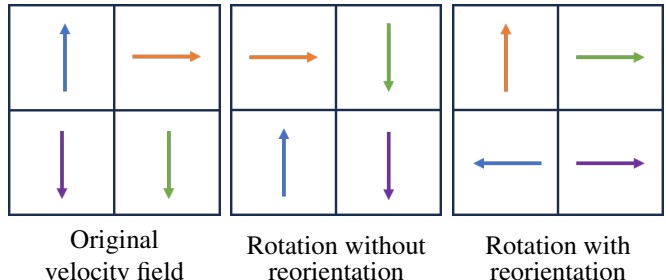

Original
velocity field

Rotation without
reorientation

Rotation with
reorientation

Figure 8: Examples of a velocity field and its augmentations with and without reorientation. Rotating by 90° counterclockwise without reorienting simply moves the spatial grid, but breaks the physical meaning of the underlying system.

**Architecture.** We use the same CNN architecture, train and evaluation setups as in Veefkind and Cesa Veefkind & Cesa (2024), which they reproduced from Wang et al. Wang et al. (2022). The architectural details can be found on the provided repository. Because the latent space has the same geometric structure as the input data, i.e. $Z = \mathbb{R}^c \times \mathbb{R}^h \times \mathbb{R}^w$ (channels×height×width), we choose a representation of $C_4$ given by the regular representation in each channel separately.

**Hyperparameters.** For both CNN models, with $C_4$ augmentations and without, and for our model, we use a learning rate of 0.001. Additionally, for our model, we set $\lambda = 0.005$. These hyperparameters were chosen after a grid search with the following values: learning rate $\in \{0.001, 0.005, 0.0001, 0.0005\}$, and equivariance coupling strength $\lambda \in \{0.005, 0.05, 0.5, 1\}$. For all other hyperparameters, we copy the values used by Veefkind and Cesa.

**Effect size analysis.** We report the effect size of our intervention in Table 15. For each model, the 'Effect' column reports the Cohen $d$-value comparing that model against 'Ours' with the regular representation. We observe that, for each model considered, there is at least one metric where the difference with our model is very large according to Cohen's $d$ statistic (Appendix G.1).

### G.5 SHREC '11 EXPERIMENT

**Data preparation.** We use the SHREC '11 dataset Lian et al. (2011); Mitchel et al. (2022) where each 3D shape is also transformed with conformal transformations. We perform augmentation according to the group $O_h$ of octahedral symmetries.

**Architecture.** We use the same architecture as the original authors Mitchel et al. (2024), which is a ResNet-based autoencoder. Similarly to the smoke experiment, the latent space retains a geometric structure. Therefore, we choose a representation of $O_h$ given by the regular representation in each channel separately.

**Hyperparameters.** Due to computational constraints, we do not perform hyperparameter tuning, and we keep the same hyperparameters as the original authors Mitchel et al. (2024), except that we set the batch size to 4. We set $\lambda = 0.5$. Additionally, we symmetrize the equivariance loss to the

Table 15: Test RMSE and effect for the SMOKE dataset. Effect values compare to 'Ours', and a positive value means ours performed better. The annotations *, **, *** indicate medium, large and very large effect sizes respectively.

| Group | Model | Future↓ | Effect | Domain↓ | Effect |
|---|---|---|---|---|---|
| N/A | CNN | 0.81 (0.01) | 3.0*** | 0.63 (0.00) | 2.8*** |
| Aug | CNN | 0.83 (0.03) | 2.2*** | 0.67 (0.06) | 1.4*** |
| N/A | MLP | 1.38 (0.06) | 14.0*** | 1.34 (0.03) | 32.6*** |
| C4 | E2CNN | 1.05 (0.06) | 6.3*** | 0.76 (0.02) | 9.5*** |
| C4 | RPP | 0.96 (0.10) | 2.5*** | 0.82 (0.01) | 21.0*** |
| C4 | Lift | 0.82 (0.01) | 4.0*** | 0.73 (0.02) | 7.6*** |
| C4 | RGroup | 0.82 (0.01) | 4.0*** | 0.73 (0.02) | 7.6*** |
| C4 | RSteer | 0.80 (0.00) | 2.8*** | 0.67 (0.01) | 6.0*** |
| C4 | PSCNN | **0.77** (0.01) | -1.0** | **0.57** (0.00) | -5.7*** |
| C4 | Ours | 0.78 (0.01) | | 0.61 (0.01) | |

decoder too, i.e., with $\lambda' = 0.8$,

$$\lambda' \, \|\rho_{\mathcal{X}}(g)(x) - D(\rho_Z(g)(E(x)))\|$$

**Effect size analysis.** We report the effect size of our intervention in Table 16. For each model, the 'Effect' column reports the Cohen $d$-value comparing that model against 'Ours' with the regular representation. We observe that, for the augmented baseline and NFT, the difference with our model is very large according to Cohen's $d$ statistic (Appendix G.1). The same analysis reveals that NIso and our model are essentially equivalent on this task.

Table 16: Test accuracies and effect for the SHREC '11 dataset. Effect values compare to 'Ours', and a positive value means ours performed better. The annotations *, **, *** indicate medium, large and very large effect sizes respectively.

| Model | Acc.↑ | Effect |
|---|---|---|
| NIso Mitchel et al. (2024) | 90.26 (1.27) | 0.1 |
| NFT Koyama et al. (2024) | 83.24 (2.03) | 3.5*** |
| AE with aug | 69.36 (2.81) | 8.5*** |
| MC Mitchel et al. (2022) | 86.5 | – |
| Ours | **90.45** (2.1) | |

# H  SENSITIVITY ANALYSIS

To assess the practical usability of our method, we performed a sensitivity analysis on the hyper-parameter $\lambda$, which controls the strength of the equivariance loss. We evaluated our model on the DDMNIST $D_4$ task across six different values for $\lambda$: $\{0, 0.05, 0.5, 1, 1.5, 2\}$, with $\lambda = 0$ being the baseline. The results, reported in Figure 9, show that while peak performance is achieved at $\lambda = 1$, the model maintains high accuracy and low variance across a wide range of values (0.5 to 2.0). This analysis demonstrates that our method is robust to the specific choice of $\lambda$.

Figure 9: Mean accuracy and standard deviation (over 5 runs) for different values of $\lambda$ on the DDMNIST $D_4$ task. $\lambda = 0$ is the baseline.

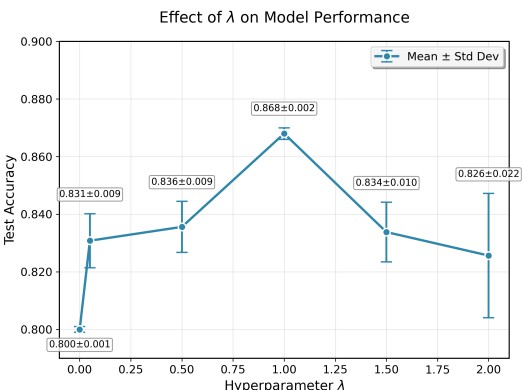

