# OpenReview forum: "An algebraic approach to approximately equivariant networks"
_ICLR.cc/2026/Conference — Submitted to ICLR 2026_

### Official Review · Reviewer_LN8t · 2025-10-21

**Soundness:** 1
**Presentation:** 1
**Contribution:** 2
**Rating:** 2
**Confidence:** 4

**Summary:**

The paper suggests training neural networks using an equivariance loss on an intermediate latent space. Given a priori knowledge of a task symmetry group, a representation of the group can be chosen and used to define the equivariance loss. Furthermore, theoretical and empirical analysis is presented that suggests that for finite groups, using multiple copies of the regular representation is the optimal choice of representation.

In experiments, the suggested approach is shown to work well.

**Strengths:**

1. The paper contributes empirical evidence that using a pre-chosen group representation in latent space and training a network to be equivariant wrt to it can work just as well as more complicated methods for obtaining (approximate) equivariance.
2. The flow from checking what representation is learned to fixing the representation is natural and well presented.

**Weaknesses:**

**Missing references:**

Encouraging equivariance by using a fixed representation in latent space is an idea that goes back at least to Cohen & Welling (Transformation Properties of Learned Visual Representations, ICLR 2015), see also Worrall & al (Interpretable transformations
with encoder-decoder networks, ICCV 2017).

Studying the case of a learnable group representation in latent space was done by Bökman & al, albeit only for keypoint description (Steerers: A framework for rotation equivariant keypoint descriptors, CVPR 2024). However, Bökman & al found that the final distribution of eigenvalues in the learned group representation depends on their initialization, which is not tested in the submitted paper.

Notably, the above works also consider infinite groups (by using finite representations), while the submitted paper states that “a direct extension to infinite groups is a non-trivial challenge for future work”.

**Theory:**

Theorem 1 contains an “almost surely”, which requires a probability measure on the space of equivariant functions considered. Such a probability measure is not presented. Even changing to “almost everywhere” or similar would require defining a measure on the function space.

Theorem 1 further only considers injective functions. Neural networks for classification are typically less and less injective the deeper layer one considers (Mahendran & Vedaldi, Understanding Deep Image Representations by Inverting Them, CVPR 2015).

**Questions:**

1. Does changing the initialization of $\rho_Z$ in Section 4.2 change the conclusions? For instance if $\rho_Z(g)$ is initialized close to the identity?
2. What layer of the network is $Z$? Is it in the middle or towards the end of the network? Do the experimental conclusions change if different layers are considered for $Z$?
3. What is meant by "almost surely" in Theorem 1? I.e. what is the measure considered?

---

> ### Author Response · Authors · 2025-11-20
> **Response to the reviewer (1/2)**
>
> We thank the reviewer for their careful reading of our submission, and for acknowledging the effectiveness of our methodology, as well as their positive comments about the presentation. We are happy to respond in detail to the queries which have been raised. We have uploaded a revised submission with changes marked in blue.
>
> • _Missing references: Encouraging equivariance by using a fixed representation in latent space is an idea that goes back at least to Cohen & Welling (Transformation Properties of Learned Visual Representations, ICLR 2015), see also Worrall & al (Interpretable transformations with encoder-decoder networks, ICCV 2017)._
>
> Thank you for these suggestions, we have added references to these works. However, our contribution is not simply the enforcement of equivariance through a loss term, which is a standard technique; rather, it is an algebraic characterization of an equivariant latent space under data augmentation. Our theorems formally motivate the choice of the regular representation, along with testable and predictive conditions for a lower bound on the number of copies of this representation (the count of linearly independent full rank embedded orbits), and our experimental results show that this choice outperforms a wide range of alternative SOTA methods for equivariant architectures, even while using a very low learnable parameter count. We believe these are novel and interesting contributions that will be of broad interest to the ICLR community.
>
> • _Studying the case of a learnable group representation in latent space was done by Bökman & al, albeit only for keypoint description (Steerers: A framework for rotation equivariant keypoint descriptors, CVPR 2024)._
>
> We appreciate this reference, and we have added a discussion in the related work. However, our work is fundamentally distinct from that of Bökman et al. Indeed, these authors only consider the groups $C_n$ and SO2 acting by rotations, and they do not give theoretical motivation for the representations that are learned. In contrast, in our paper, we take a more general approach:
>
> 1) Give a rigorous mathematical explanation, together with testable and predictive conditions, for why the regular representation in particular emerges.
> 2) Extend the analysis to other groups, including, crucially, non-abelian groups (e.g. $D_3$);
> 3) Consider groups that act non-geometrically (e.g. $C_2$ and font swaps), which are unrelated to “keypoint descriptions”, as acknowledged by the reviewer.
>
> Finally, we agree with the reviewer that our work is not the first to study learned group representations in the latent space, but we do not believe this to be our novel contribution (see response to previous point).
>
> • _However, Bökman & al found that the final distribution of eigenvalues in the learned group representation depends on their initialization, which is not tested in the submitted paper._
>
> We thank the reviewer for raising the point about initialization. For this reason, we have added a section (Appendix F.5) that discusses different initialization schemes. The main take-away is that constrained initialisation schemes hinder the ability of the model to find high-performing representations, and we show that our theory applies in this setup too (Appendix F.5). We discuss this further in an additional comment below.
>
> • _Notably, the above works also consider infinite groups (by using finite representations), while the submitted paper states that “a direct extension to infinite groups is a non-trivial challenge for future work”._
>
> We thank the reviewer for raising this point. Our method does apply to infinite groups by selecting a rich finite subgroup, and using its regular representation. The quoted sentence was not well-phrased, and we have updated that discussion in the paper (Section 7, Limitations and future work). Indeed, we exhaustively benchmark our approach against other models such as NIso, SCNN, RPP, MC, NFT and PSCNN which directly support continuous symmetries such as SO(3), O(3) and the conformal group (Tables 4 and 5(b)), and show that our method consistently outperforms them in the vast majority of cases.
>
> While the cited work uses a finite representation for the infinite groups O(n) and SO(n), it is not an extension of our method. Bokman et al., employ the same strategy as Weiler and Cesa, 2019, https://arxiv.org/abs/1911.08251, to handle the case of O($n$) and SO($n$), which is to truncate the representation of the corresponding group after a fixed number of irreps to obtain a finite representation. Crucially, this method does not yield a regular representation of the group or a subgroup (our method), and hence is not an extension of our method, even for the groups O($n$) and SO($n$).

---

> > ### Author Response · Authors · 2025-11-20
> > **Response to the reviewer (2/2)**
> >
> > • _Theory: Theorem 1 contains an “almost surely”, which requires a probability measure on the space of equivariant functions considered. Such a probability measure is not presented. Even changing to “almost everywhere” or similar would require defining a measure on the function space ...
> > What is meant by "almost surely" in Theorem 1? I.e. what is the measure considered?_
> >
> > We have comprehensively rigorized the proof of Theorem 1, which has been split into two statements (Theorem 1 and 2), applying results from measure theory of real analytic functions. In particular, we make explicit all the probability measures involved, which are to be taken on the parameter space $\Theta \subseteq \mathbb R^p$. We believe this fully addresses the reviewer’s concerns regarding measure-theoretic aspects.
> >
> > • _Theorem 1 further only considers injective functions. Neural networks for classification are typically less and less injective the deeper layer one considers (Mahendran & Vedaldi, Understanding Deep Image Representations by Inverting Them, CVPR 2015)._
> >
> > We are grateful for this observation. Our refactored proofs do not require injectivity. We hope this fully addresses the reviewer’s concerns on this point.
> >
> > • _Does changing the initialization of $\rho_Z$ in Section 4.2 change the conclusions? For instance if $\rho_Z(g)$ is initialized close to the identity?_
> >
> > We thank the reviewer for their comments, and have added an initialization sensitivity analysis (see Section 4.2 and Appendix F.5). In summary:
> > 1) What does NOT change. Our main claim in Section 4.2 still holds: each linearly independent full rank embedded orbit corresponds to a distinct copy of the regular representation, even when the representation is initialized close to the identity. We have added Appendix F.5 to show these results.
> > 2) What does change. When $\rho_Z$ is initialized close to the identity, the network is biased towards an over-representation of the trivial representation, as this strong priming constrains its parameter search. This is supported by an additional ablation study with the trivial representation in the main experiments (Tables 3, 4), where we show that this always underperforms compared to the regular representation. We discuss this phenomenon in Appendix F.5. To make this clear in the main body, we have added a specification in Section 4.2 (Line 237-241)  where we explicitly say that we initialize the representation randomly from $\mathcal N(0,1)$, and discuss other initializations in Appendix F.5.
> >
> > • _What layer of the network is $Z$? Is it in the middle or towards the end of the network? Do the experimental conclusions change if different layers are considered for $Z$?_
> >
> > If different layers are considered for $Z$, our experimental conclusions do not change. To investigate this, we have now explicitly added a further ablation study in Appendix F.4.
> >
> > For the main experiments we choose $Z$ to be the layer expected to hold the most significant latent encodings. For example, for an encoder/decoder architecture (Smoke, SHREC ‘11) we choose the middle layer; for a classifier (DDMNIST, MedMNIST3D) we choose the penultimate layer before the final classifier head. This is now explicit in the revised manuscript, in the captions of Tables 1 and 2 for Section 4.2, and in lines 364-366 within Section 6.
> >
> > We thank the reviewer again for their careful comments and look forward to their response.

---

> ### Comment · Reviewer_LN8t · 2025-11-21
>
> I appreciate the well-formulated response and updated manuscript.
>
> **Regarding the related work:** I believe the reworking of the related work section to be adequate.
>
> **Theorem 1:** Theorem 1 is believable since it now conditions the existence of a regular representation on the latent space both on existence of any linear representation on the latent space and the fact that one data point orbit is faithfully reproduced in the latent space. But, to my understanding, this does not tell us what the "best" representation on the latent space for a given task is. For instance, for invariant tasks, if we take $Z$ to be the final output space, we want collapse of the data point orbits to happen.
>
> **Theorem 2:**
>
> 1. The statement of Thm 2 is unclear to me. As far as I understand, the statement does not depend on the data or loss function, so we are free to choose those arbitrarily. Hence we can choose the loss function to enforce the trivial representation on the output of $E$. Note that we can still choose an architecture that is capable of encoding multiples of the regular representation as the representation acting on the output, so that case (i) of the Theorem does not apply. So we get to case (ii) which says that the output contains a copy of the regular representation with probability 1. But the network is trained to output to the trivial representation so this seems contradictory.
> 2. Comment 1 also relates to the new experiments in F.4 and F.5 where choices of layer $Z$ and initialization of $\rho_Z$ are ablated. I appreciate these experiments and think that they strengthen the paper. Here the chosen task has an output that contains the regular representation. But what would happen if the output is invariant? E.g. what happens on the classification task on CIFAR10? If $Z$ is among the last layers, intuitively it would be strange if there was a copy of the regular representation left in latent space.
> 3. A further potential problem with the probabilistic setup is the following. At initialization, the network is not linearly equivariant. In the proposed probabilistic framework, what is the probability that the network is equivariant after training?

---

> > ### Author Response · Authors · 2025-11-24
> > **Response to the reviewer's comment (1/2)**
> >
> > We thank the reviewer for engaging with the new version of the manuscript and for their further comments.
> >
> > _Theorem 1 is believable since it now conditions the existence of a regular representation on the latent space both on existence of any linear representation on the latent space and the fact that one data point orbit is faithfully reproduced in the latent space. But, to my understanding, this does not tell us what the "best" representation on the latent space for a given task is. For instance, for invariant tasks, if we take Z to be the final output space, we want collapse of the data point orbits to happen._
> >
> > Thanks for these comments. While we agree with the reviewer that Theorem 1 does not imply that the regular representation will be optimal for a particular task, the perspective we take in this paper is that this question is best answered empirically by downstream task performance. For this reason, in Section 6, we provide ablations with the trivial and defining representations, which consistently show that the regular representation has the best performance.
> >
> > For invariant tasks, we certainly agree that the data orbits must collapse on the final layer, or output space $\mathcal{Y}$ (i.e. the action $\alpha_\mathcal Y$, in our paper’s notation, must be trivial). However, our results on invariant classification tasks (Tables 3 and 4) show that, even when $Z$ is chosen as the penultimate layer (i.e. the final hidden layer), the regular representation is in fact optimal.
> >
> > _1. The statement of Thm 2 is unclear to me. As far as I understand, the statement does not depend on the data or loss function, so we are free to choose those arbitrarily. Hence we can choose the loss function to enforce the trivial representation on the output of E. Note that we can still choose an architecture that is capable of encoding multiples of the regular representation as the representation acting on the output, so that case (i) of the Theorem does not apply. So we get to case (ii) which says that the output contains a copy of the regular representation with probability 1. But the network is trained to output to the trivial representation so this seems contradictory._
> >
> > We thank the reviewer for this interesting scenario. From our understanding, the reviewer is enquiring about the setup where gradient descent yields an exactly equivariant encoder and an exactly trivial representation. Our theorem proves that either (i) the architecture is not capable of encoding the regular representation, or (ii) the regular representation appears with probability 1. Hence, we argue that the setup proposed by the reviewer, where gradient descent yields a function equivariant to the trivial representation, must in fact arise from case (i), or else it has probability 0.
> >
> > Since the theorem holds, one of the assumptions must therefore be false. In particular, we conclude that for gradient descent to learn an exactly equivariant encoder with respect to the trivial representation, when the encoder has sufficiently expressive structure, is a probability-0 event. We believe that this thought experiment, despite being perhaps counterintuitive, demonstrates the power of our result in an interesting way.

---

> ### Author Response · Authors · 2025-11-24
> **Response to the reviewer's comment (2/2)**
>
> _2. Comment 1 also relates to the new experiments in F.4 and F.5 where choices of layer $Z$ and initialization of $\rho_Z$ are ablated. I appreciate these experiments and think that they strengthen the paper. Here the chosen task has an output that contains the regular representation. But what would happen if the output is invariant? E.g. what happens on the classification task on CIFAR10? If $Z$ is among the last layers, intuitively it would be strange if there was a copy of the regular representation left in latent space._
>
> We agree with the reviewer that for an invariant classification task, we might naively expect that the trivial representation would be optimal at the penultimate layer (i.e. the final hidden layer), since in the subsequent layer (i.e. the output layer) invariant behaviour is required. However, this is not what our CIFAR10 results show (Table 2): in such a configuration, for the penultimate layer, the network learns a copy of the regular representation. Even stronger evidence for the optimality of this representation is shown in Tables 3 and 4, which are also invariant classification tasks, where in both cases we see that the regular representation outperforms the defining and trivial representations, again on the penultimate layer. We believe these surprising results are strengths of our work, which will interest a wide audience.
>
> In a further revision to the manuscript, we have drawn attention to this with a new bullet point at the end of the introduction: “Our experiments show that the regular representation is optimal across a wide range of diverse learning tasks, even in conditions where this may be surprising, such as in the penultimate layer of an invariant classification task”.
>
> _3. A further potential problem with the probabilistic setup is the following. At initialization, the network is not linearly equivariant. In the proposed probabilistic framework, what is the probability that the network is equivariant after training?_
>
> We thank the reviewer for raising this point. Computing the probability that GD will yield such a function is potentially intractable. However, in a ML setup, we interpret Theorem 2 in a motivational way for our choice of fixing the regular representation. Our suggested reading of this result is the one in the “Key Theoretical Insight” box, that is: “To achieve encoder equivariance”. In other words, except for inexpressive encoders, the regular representation is a requirement for exact equivariance.
>
> Theorem 2 is simply a guarantee on the invertibility of the matrix $M$, and a similar result would hold for non-equivariant encoders. However, the invertibility of this matrix is significant in a representation-theoretic sense only when $E$ is equivariant (the setup of Theorem 1), which is why we ask for gradient descent to yield an equivariant function.

---

> > ### Comment · Reviewer_LN8t · 2025-11-25
> >
> > I thank the authors for the response.
> >
> > ### *Since the theorem holds, one of the assumptions must therefore be false.*
> >
> > The theorem can also be false. Before reviewing the proof I would like to clearly understand the statement.
> >
> > ### *While we agree with the reviewer that Theorem 1 does not imply that the regular representation will be optimal for a particular task, the perspective we take in this paper is that this question is best answered empirically by downstream task performance.*
> >
> > This is a valid perspective, and the same perspective prior work has used. It does however somewhat contradict previous statements such as "*In contrast, in our paper, we take a more general approach: 1. Give a rigorous mathematical explanation, together with testable and predictive conditions, for why the regular representation in particular emerges.*"
> >
> > ### *We agree with the reviewer that for an invariant classification task, we might naively expect that the trivial representation would be optimal at the penultimate layer (i.e. the final hidden layer), since in the subsequent layer (i.e. the output layer) invariant behaviour is required.*
> >
> > I could not find the exact architecture used. Is the "classifier head" just a single linear layer? Is there average pooling before that? Why does the theorem not hold if we take $Z$ to be the final output that should be invariant?
> >
> > ### *Even stronger evidence for the optimality of this representation is shown in Tables 3 and 4, which are also invariant classification tasks, where in both cases we see that the regular representation outperforms the defining and trivial representations, again on the penultimate layer.*
> >
> > In these experiments, where you enforce the trivial representation through a loss, does the network still learn a copy of the regular representation? That seems to be the prediction from Thm 2.
> >
> > You have found that using multiple copies of the regular representation outperforms some other choices of representations. This is a nice empirical finding, but does not show that there is not another choice that is even better.
> >
> > **Regarding the probabilistic setup**
> >
> > If the probability of the network being equivariant after training is 0, then Thm 2 seems to lose practical relevance (and might also have theoretical problems due to conditioning on a null event). For instance, when training with a learnable $\rho$, it seems to me like the probability of $\rho$ hitting the measure 0 subset of matrices satisfying the algebraic constraints for a group representation is 0 in the proposed setup.

---

> > > ### Author Response · Authors · 2025-11-27
> > > **Response to the reviewer's comment (1/2)**
> > >
> > > We thank the reviewer for their further comments. We are happy to respond in detail here:
> > >
> > > • _This is a valid perspective, and the same perspective prior work has used. It does however somewhat contradict previous statements such as "In contrast, in our paper, we take a more general approach: 1. Give a rigorous mathematical explanation, together with testable and predictive conditions, for why the regular representation in particular emerges."_
> > >
> > > We believe there is no contradiction here. The theory and the experiments answer two different questions, which we summarize as: (1) “Why does the regular representation emerge?”, answered by the theory (Theorem 1 and 2); and (2) “What is the optimal representation for a downstream task?”, answered by the experiments in Section 6.
> > >
> > > • _I could not find the exact architecture used. Is the "classifier head" just a single linear layer? Is there average pooling before that?_
> > >
> > > Yes, the classifier head is just a single linear layer. This is defined in our code. There is no average pooling before that.
> > >
> > > • _Why does the theorem not hold if we take $Z$ to be the final output that should be invariant?_
> > > If we have exact equivariance to the trivial representation, then the theorem holds, and case (ii) is impossible since the representation does not contain the regular representation. Therefore, we must be in case (i).
> > > If we have approximate equivariance to the trivial representation, then the theorem does not hold, since it is a theorem about exactly equivariant encoders. In this case, one could imagine a more general theorem about approximately-equivariant encoders, in which case (i) would be adapted to include the case of a very small determinant. This is what one would encounter in practice when training an approximately equivariant encoder.
> > > We believe that this does not undermine the value of our theorem, which is to give a principled mathematical explanation for the presence of the regular representation in the output space of an equivariant encoder, which certainly has gaps with respect to application to realistic experimental scenarios, but which is then strongly validated by our experiments in Sections 4 and 6. As mentioned above, one must also keep in mind that we are operating in reality with machine-precision arithmetic, and real experiments are never perfectly described by any particular analytical result.
> > >
> > > • _In these experiments, where you enforce the trivial representation through a loss, does the network still learn a copy of the regular representation? That seems to be the prediction from Thm 2._
> > >
> > > In these experiments (Tables 3 and 4), as in all experiments from Section 6, we fix the regular representation, according to our method described in Section 5.
> > >
> > > • _You have found that using multiple copies of the regular representation outperforms some other choices of representations. This is a nice empirical finding, but does not show that there is not another choice that is even better._
> > >
> > > We have always been careful with our wording here; for example we have said that our study “provides evidence” for optimality of the regular representation, and in our abstract we state that “the regular representation … outperforms defining and trivial representation baselines”. We do not claim to present a deductive proof that the regular representation is in fact optimal. If we agree that the question of optimality is ultimately best answered empirically, we must accept this limitation as for any empirical study.
> > > We were careful in our choice of ablation studies, where we compare the regular representation with both the trivial representation, a standard intuitive choice for invariant tasks, and the defining representation, a rich and geometrically meaningful representation of the group. In both cases, we find that the regular representation overperforms (Tables 3 and 4). Indeed, across all the experiments we have conducted for this research project, we have never found a scenario where some other representation gives a better performance for a particular task than the regular representation.

---

> > > > ### Author Response · Authors · 2025-11-27
> > > > **Response to the reviewer's comment (2/2)**
> > > >
> > > > • _If the probability of the network being equivariant after training is 0, then Thm 2 seems to lose practical relevance (and might also have theoretical problems due to conditioning on a null event). For instance, when training with a learnable $\rho$, it seems to me like the probability of  $\rho$ hitting the measure 0 subset of matrices satisfying the algebraic constraints for a group representation is 0 in the proposed setup._
> > > >
> > > > The reviewer is correct in saying that it is highly unlikely that gradient descent will yield a perfect representation.
> > > > Nonetheless, in the experiments in Section 4, we were able to learn a $\rho$ with errors as small as 1E-10. Of course, in any machine-precision setup, we must accept some degree of deviation from ideal conditions.
> > > > However, we remark that we primarily use Theorem 2 to motivate, in a principled way, our choice to fix the regular representation – a choice empirically supported by the experiments in Section 4.
> > > >
> > > >
> > > > We hope that these responses address the reviewer's concerns, and we are happy to engage in further discussion.

---

> > > > > ### Comment · Reviewer_LN8t · 2025-11-27
> > > > >
> > > > > I would like to say that I sincerely appreciate the willingness of the authors to discuss.
> > > > >
> > > > > There are however still multiple points of contention.
> > > > >
> > > > > #### *The theory and the experiments answer two different questions, which we summarize as: (1) “Why does the regular representation emerge?”, answered by the theory (Theorem 1 and 2)*
> > > > >
> > > > > It is in my view a stretch to say that the theory answers this question if the theory is not applicable to the training of neural networks or even some idealized version of training neural networks.
> > > > >
> > > > > #### *Yes, the classifier head is just a single linear layer. This is defined in our code. There is no average pooling before that.*
> > > > >
> > > > > Thank you for confirming. The answer makes it sound like there is code available to review, have I missed that?
> > > > >
> > > > > #### *If we have exact equivariance to the trivial representation, then the theorem holds, and case (ii) is impossible since the representation does not contain the regular representation. Therefore, we must be in case (i).*
> > > > >
> > > > > I can easily write down a neural network that can both encode the trivial representation on the output and the regular. For instance a single linear layer from 2 dimensions to 2 dimensions, with the action on the input being permutation. If the layer has the form $x\mapsto \begin{pmatrix}a & a \\\\ b & b\end{pmatrix}x$ then it is invariant, if it is of the form $\begin{pmatrix}a & b \\\\ b & a\end{pmatrix}$, then it is permutation equivariant. Hence case (i) is not relevant here. Which of these two forms the layer learns must be dependent on the loss, but the loss is missing from the theory. You can certainly get out of this conundrum by saying that the theory never applies, but then I do not see why the theory is interesting.
> > > > >
> > > > > #### *[...] one could imagine a more general theorem about approximately-equivariant encoders, in which case (i) would be adapted to include the case of a very small determinant. This is what one would encounter in practice when training an approximately equivariant encoder.*
> > > > >
> > > > > I mostly agree with this. Instead of adjusting case (i) I would expect that the network could be approximately equivariant with respect to more than one representation. E.g. you can have full rank of $E(\mathcal{O}_x)$ while still having that the orbit is very close to collapsed so that the trivial representation more accurately describes the situation than the regular. And this could happen in a network where not all instances of the parameters lead to almost collapsed orbits.
> > > > >
> > > > > #### *In these experiments (Tables 3 and 4), as in all experiments from Section 6, we fix the regular representation, according to our method described in Section 5.*
> > > > >
> > > > > In tables 3 and 4 you have rows saying "Trivial rep", are these not trained with a loss to encourage the trivial representation? What I'm asking is whether the theorem applies to that case or not and why.
> > > > >
> > > > > **Summary of my current thoughts about the paper**
> > > > >
> > > > > If you had theory motivating the use of the regular representation in all layers for all tasks, then I would agree that this is novel and an extremely strong theoretical contribution. But this is so strong that I believe it to be impossible to be true, see the example with $2\times 2$ matrices above. On the other hand, if you say that the theory is not actually applicable, this novelty disappears and you cannot say that you have a more principled approach than prior work as it is not novel to impose a pre-defined representation on the latent space. If this paper was more clear about the fact that it presents an empirical study and also more clearly related this empirical study to prior work I would have few qualms about giving a score 6. I do however believe that this would require a significant rewrite and much more downweighted attributions to the theory.

---

### Official Review · Reviewer_g1LT · 2025-10-28

**Soundness:** 3
**Presentation:** 2
**Contribution:** 2
**Rating:** 2
**Confidence:** 3

**Summary:**

The authors consider the problem of learning network that respect an equivariance condition. They argue for not confining the architecture to be exactly equivariant, but rather to penalize non-adherence to the equivariance conditions.

A problem with imposing equivariance by architecturial means in general is the a-priori non-fixed action of the group on the intermediate spaces. The authors argue theoretically as well as empirically that the regular action of the group is a reasonable choice. Using this choice, the authors propose to train an architecture consisting of an encoder E to a latent space Z (on which the group is acting regularly) and then a decoder/classifier/etc. D from Z to the output space, regularizing E to be equivariant with respect to the fixed input action and regular action on Z during training.

**Strengths:**

The manuscript is easy to read. The main ideas are easy to digest. The main idea, to achieve equivariance through regularization rather than hard constraints, is sound.

**Weaknesses:**

* Clarity *  While the manuscript as a whole is simple to read, there are many details that remain unclear.

1. The statement and proof of the main theorem is correct, but I am unsure of its significance. It is only applicable to quite small groups because of the condition $\dim(Z)\geq \abs{G}$ (think about the group of permutation on, say, more than 10 elements). If the action $\alpha_X$ is linear, one can always $\rho_Z=\alpha_X$ and $\calA = X$ and choose $E$ as the identity to obtain an injective, equivariant map, irregardless of whether $X$ contains the regular representation.

Also, the interpretation the authors make (the 'key theoretical insight' on page 4) is unclear to me. First, it is unclear what the meaning of a 'linearly independent orbit' is. Does it mean that the spans of the images of the two orbits have a trivial intersection? That they are not contained in each other? It is also unclear how the statement follows from Theorem 1 : While the group acts transitively on each orbit, it does not act transitively on the union of the orbits.

2. In the exploratory experiments, the authors seem to be making a point out of that the irrep counts always exactly corresponds to the number of linearly independent embedded orbits. It is however unclear how this number is measured, also after looking at the code.

3. The authors claim in the introduction that their method leads to lower parameter counts. I do not understand this. In fact, a constrained architecture will in some sense always have a lower number of effective parameters -- the dimension is reduced by the restriction.

See questions on smaller details below.

* Novelty * What the authors ultimately propose is to penalize non-conformance to the equivariance condition both for the network as a whole and the encoder part. This ultimately is very close to simply using augmentations, and is the driving idea behind e.g. residual pathway priors (see also [1] and references therein). If the only novel idea is the use of the regular representation on the latent space, it is limited.

* Experimental validation* The authors test their method in three settings, and they do showcase good performance. However, their method never outperforms their baseline by more than a standard deviation over three runs, which is slightly unconvincing.


[1] Pertigkiozoglou, S., Chatzipantazis, E., Trivedi, S., & Daniilidis, K. (2024). Improving equivariant model training via constraint relaxation. Advances in Neural Information Processing Systems, 37, 83497-83520.

**Questions:**

See the questions under weaknesses. Consider also the following more detailed questions below:

1. In the main body of the text, the TMNIST experiments use a latent dimension of 8, but in the appendix and the code supplement, a latent dimension of 6 is claimed. Which is correct?

2. For the MNIST/D3 experiments, the authors say that the digits are augmented by 'arbitrary' rotations. Does this mean that the rotations are random $\mathrm{SO}(2)$-rotations, or something else?  The group is then chosen as $D_3$, which also contains reflections. Can the authors comment why this choice is made (it does not seem to be compatible with the symmetries of the dataset?)

3. In the main text, the rotations building D3 are 120 degrees, but in the appendix, they are specified as rotations of 60 degrees. Which is correct?

4. In Table 4, the wrong number in the Nodule column seems to have been underlined.

5. In the appendix, the inclusion of Figure 7 is confusing,  in that it is never referenced. In which experiment is reorientation applied/not applied?

---

> ### Author Response · Authors · 2025-11-20
> **Response to the reviewer (1/3)**
>
> We thank the reviewer for their comprehensive comments and for acknowledging the clarity and soundness of our work. We are happy to respond in detail here. We have uploaded a revised submission with changes marked in blue.
>
> • _the main theorem is correct, but I am unsure of its significance. It is only applicable to quite small groups because of the condition $\text{dim}(Z) \geq |G| $ (think about the group of permutation on, say, more than 10 elements)._
>
> Thank you for this question. The finite groups that we study in our work are very prominent in the equivariant ML/GDL community, and are often used by other equivariant methodologies, even for those that apply to infinite groups. We report a non-exhaustive table containing recent literature, peer-reviewed at top-tier ML venues, containing experiments with the groups we considered in our work:
>
> $C_2$
> Weiler & Cesa, NeurIPS 2019, https://arxiv.org/abs/1911.08251
> Finzi et al., NeurIPS 2021, https://arxiv.org/abs/2112.01388
> Maile et al., ICLR 2023, https://arxiv.org/abs/2210.05484
> Zhao et al., ICLR 2023, https://arxiv.org/abs/2210.05484
> Wu et al., ICLR 2025, https://arxiv.org/abs/2408.11760
> Wang et al., ICML 2025, https://arxiv.org/abs/2310.02299
> Veefkind and Cesa, ICML 2024, https://arxiv.org/abs/2406.03946
>
> $C_4$
> Cohen & Welling, ICML 2016, https://arxiv.org/abs/1602.07576
> Weiler & Cesa, NeurIPS 2019, https://arxiv.org/abs/1911.08251
> Finzi et al., NeurIPS 2021, https://arxiv.org/abs/2112.01388
> Wang et al., ICML 2022, https://arxiv.org/abs/2201.11969
> Maile et al., ICLR 2023, https://arxiv.org/abs/2210.05484
> Zhao et al., ICLR 2023, https://arxiv.org/abs/2210.05484
> Wu et al., ICLR 2025, https://arxiv.org/abs/2408.11760
> Wang et al., ICML 2025, https://arxiv.org/abs/2310.02299
> Veefkind and Cesa, ICML 2024, https://arxiv.org/abs/2406.03946
> Pertigkiozoglou et al., NeurIPS 2024, https://arxiv.org/abs/2408.13242
>
> $D_4$
> Cohen & Welling, ICLR 2017, https://arxiv.org/abs/1612.08498
> Weiler & Cesa, NeurIPS 2019, https://arxiv.org/abs/1911.08251
> Finzi et al., NeurIPS 2021, https://arxiv.org/abs/2112.01388
> Zhao et al., ICLR 2023, https://arxiv.org/abs/2210.05484
> Veefkind and Cesa, ICML 2024, https://arxiv.org/abs/2406.03946
> Zhang et al., NeurIPS 2024, https://openreview.net/pdf?id=HRnSVflpgt
>
> We emphasise that we have extensively benchmarked our method against continuous groups, namely SO(3), O(3) and the conformal group, handling these cases by choosing appropriate large finite subgroups, i.e. $\text{Sym}_\text{Cube}$ (order 24) and $O_h$ (order 48). We show in the majority of these cases that our method is comparable with or outperforms other heavily-parameterised methods that are specialised to continuous groups (Tables 4 and 5(b)).
>
> However, we agree that the case of very large finite groups, e.g. the full permutation groups Sym($N$) for N>10, are interesting to consider. In this case, we propose that the best way to apply our method would be to choose a suitable subgroup $G$ of Sym($N$) with size dividing the dimension of the latent space, and apply our method to $G$. This is indeed the same strategy we successfully employed for infinite groups.
>
> • _If the action $\alpha_X$ is linear, one can always  $\rho_Z=\alpha_X$ and  $A=X$ and choose $E$ as the identity to obtain an injective, equivariant map, irregardless of whether $X$ contains the regular representation._
>
> If our other conditions are met, namely that $X$ contains a training element $x$ with a free and transitive action of $G$ such that the orbit $E(O_x)$ is full rank, then under this scenario of an identity encoder our theorem will hold, and latent space $Z$ will indeed contain a copy of the regular representation according to this orbit (Theorem 1). We do not feel that this example contradicts our result as stated in the submitted or updated version, although we would be very happy to discuss it more fully if the reviewer could explain further their concerns.
>
> However, we note that in our revised version of the article, we have significantly revised the statement and proof of the theorem. In particular, the revised theorem no longer requires an injectivity condition.
>
> • _Also, the interpretation the authors make (the 'key theoretical insight' on page 4) is unclear to me. First, it is unclear what the meaning of a 'linearly independent orbit' is. Does it mean that the spans of the images of the two orbits have a trivial intersection? That they are not contained in each other?_
>
> This is correct, here we mean exactly that the spans of the images of the orbits have trivial intersection. We are grateful for this point and have clarified this in the manuscript (Appendix E, Definition 4).

---

> > ### Author Response · Authors · 2025-11-20
> > **Response to the reviewer (2/3)**
> >
> > • _It is also unclear how the statement follows from Theorem 1 : While the group acts transitively on each orbit, it does not act transitively on the union of the orbits._
> >
> > We do not require the group to act transitively on the union of the orbits. Our claim is simply that if we can find two independent representations (in our case, regular representations) $Z_1$ and $Z_2$ in $Z$, then if they are linearly independent, we conclude that $Z$ contains the direct sum representation $Z_1 \oplus Z_2$. That is to say, $Z$ contains two separate copies of the regular representation, one due to $Z_1$ and the other due to $Z_2$. This is now exemplified in Figure 2.
> >
> > • _In the exploratory experiments, the authors seem to be making a point out of that the irrep counts always exactly corresponds to the number of linearly independent embedded orbits. It is however unclear how this number is measured, also after looking at the code._
> >
> > We thank the reviewer for raising this point, and we are happy to clarify. An embedded orbit $E(O_x)$ is constructed by first embedding a data point $x \mapsto E(x)$, and then constructing the matrix with columns given by $\widehat \rho_Z(g_i)(E(x))$ for all $i$. Finally, we check that two embedded orbits $E(O_x)$, $E(O_x’)$ are linearly independent by checking whether the concatenated matrix is of full rank (i.e. if all singular values are non-zero). This same methodology is extended to any finite number n of orbits by simply concatenating the n corresponding matrices and calculating their rank. We have clarified this methodological point in the paper, in Appendix F.1 and Section 4.2  in the revised manuscript.
> >
> > • _The authors claim in the introduction that their method leads to lower parameter counts. I do not understand this. In fact, a constrained architecture will in some sense always have a lower number of effective parameters -- the dimension is reduced by the restriction._
> >
> > This is a good point and we have now clarified this in Sections 1 and 2. The reviewer is correct that equivariant architectures may be able to reduce their parameter counts due to weight-tying, however in practice these comparison architectures often require greatly increased numbers of parameters to achieve good performance, as shown in the tables in Section 6. For example, for the DDMINST D4 experiment (Table 3), the RPP and PSCNN models require 1.73E6 and 1.23E6 parameters respectively to achieve the reported performance (hyperparameter-tuned by the original authors), while our architecture requires only 3E4 parameters, the same as the baseline CNN. As a result, our model requires considerably reduced computational effort to train. Nonetheless, our method strongly outperforms baseline, and exceeds or is competitive with the highest-performing alternative architectures.
> >
> > • _Novelty * What the authors ultimately propose is to penalize non-conformance to the equivariance condition both for the network as a whole and the encoder part. This ultimately is very close to simply using augmentations, and is the driving idea behind e.g. residual pathway priors (see also [1] and references therein). If the only novel idea is the use of the regular representation on the latent space, it is limited._
> >
> > We would like to emphasize that our contribution is not simply the enforcement of equivariance through a loss term. Rather, it is an algebraic characterization of an equivariant latent space under data augmentation. Our theorems formally motivate the choice of the regular representation, along with testable and predictive conditions for a lower bound on the number of copies of this representation (the count of linearly independent full rank embedded orbits). Another principal contribution is to show that this low-parameter approach can outperform or match a wide range of sophisticated SOTA equivariant architectures. We believe these are novel and interesting contributions that will be of broad interest to the ICLR community, and go far beyond the simple adoption of the regular representation on the latent space.
> >
> > The reviewer is correct that our method is intimately linked to augmentations. For this reason we have been careful to include an augmented baseline in all our experiments. Nonetheless, we have shown that our method consistently outperforms this augmented baseline, with a large effect size. We also show our model consistently outperforms the Residual Pathway Priors architecture (RPP in our results tables) with a large effect size, for all runs except one.

---

> > > ### Author Response · Authors · 2025-11-20
> > > **Response to the reviewer (3/3)**
> > >
> > > • _Experimental validation* The authors test their method in three settings, and they do showcase good performance. However, their method never outperforms their baseline by more than a standard deviation over three runs, which is slightly unconvincing._
> > >
> > > To address this concern, we measure the effect size of our intervention with Cohen’s d statistic, which measures the difference between two distributions in pooled standard deviation units. This analysis reveals that, in the majority of cases, our methodology has a large to very large effect, and in particular, outperforms every CNN baseline or alternative model on the majority of datasets by at least one standard deviation (except for PSCNN in Smoke, and NIso in SHREC). We have added a discussion of Cohen’s d statistic in Appendix G.1, and we report effect size analysis for each experiment in the corresponding appendix (G.2, G.3, G.4, G.5). Finally, we remark that Cohen’s d statistic is a widely used metric to assess effect size in the machine learning literature (e.g. Miranda et al., 2025; Huang et al., 2025; Gundersen et al., 2023; Karandikar et al., 2021; Hermann et al., 2024; these citations are reported in Appendix G.1).
> > >
> > > • _In the main body of the text, the TMNIST experiments use a latent dimension of 8, but in the appendix and the code supplement, a latent dimension of 6 is claimed. Which is correct?_
> > >
> > > Thank you for spotting this: 8 is the correct one, as stated in the main text. We have updated the appendix and will update the code in due course.
> > >
> > > • _For the MNIST/D3 experiments, the authors say that the digits are augmented by 'arbitrary' rotations. Does this mean that the rotations are random rotations, or something else? The group is then chosen as $D_3$, which also contains reflections. Can the authors comment why this choice is made (it does not seem to be compatible with the symmetries of the dataset?)_
> > >
> > > This was a deliberate choice to show that the underlying dataset augmentations do not need to coincide with the equivariance enforced. The cleanest way to see this is to consider the augmented dataset as the dataset itself, and on that dataset, one can enforce equivariance with a chosen group action.
> > >
> > > To be even more explicit, in $L_\text{opt}$ (beginning of Section 4.2), the $x_i$ can undergo any transformation (such as random rotations), because we can construct a new dataset $\mathcal X’$ containing all augmented versions from $\mathcal X$ according to a group $H$. However, the $\alpha_\mathcal X$, $\alpha_ \mathcal Y$ and $\rho_Z$ are actions/representations of a potentially different group $G$, which is $D_3$ in the case raised by the reviewer.
> > >
> > > • _In the main text, the rotations building D3 are 120 degrees, but in the appendix, they are specified as rotations of 60 degrees. Which is correct?_
> > >
> > > Thank you for spotting this. 120 degrees is the correct one, and we have corrected the appendix.
> > >
> > > • _In Table 4, the wrong number in the Nodule column seems to have been underlined._
> > >
> > > Thank you for spotting this. This has been rectified.
> > >
> > > • _In the appendix, the inclusion of Figure 7 is confusing, in that it is never referenced. In which experiment is reorientation applied/not applied?_
> > >
> > > It refers to the SMOKE experiment, and it is now appropriately referenced. This picture argues why reorientation is fundamental in the SMOKE experiment, as not applying it would make the underlying physics meaningless.
> > >
> > > We thank the reviewer again for their careful comments and look forward to their response.

---

> > > > ### Comment · Reviewer_g1LT · 2025-11-25
> > > > **Response**
> > > >
> > > > I thank the authors for the detailed response. All of my minor issues have been resolved. I in particularly welcome the more transparent reports about how data in the tables in the exploratory experiments has been measured, and acknowledge the added statistical measure of the performance increase.
> > > >
> > > > I remain a bit skeptical of the claimed contribution and its practical relevance. Reviewer LN8t below makes a very good point about the updated Theorem 2: It seems to make a statement about a probability zero event - if the network is initialized not equivariant, it will at least intuitively to me, by the argument of the theorem, with probability one remain not equivariant, meaning that it will never be in a position where Theorem 1 is applicable.
> > > >
> > > > The discussion has also made what conclusions should be drawn from the theoretical results unclear to me. The authors both say that the regular representation on the latent space is necessary to achieve equivariance, but then (again in the discussion with reviewer LN8t) claim that "Theorem 1 does not imply that the regular representation will be optimal for a particular task, the perspective we take in this paper is that this question is best answered empirically by downstream task performance."
> > > >
> > > > In the updated manuscript, the authors still claim that they present a "novel parameter-free method to improve performance". If this novel method is to fix the regular representation in the latent space and then penalize the encoder from not being equivariant towards that, I still believe that this is not novel.
> > > >
> > > > Given the fact that the details of the theorem is still unclear to me, and that I feel that the contribution beyond that is limited, I will retain my score.

---

### Official Review · Reviewer_awTB · 2025-11-04

**Soundness:** 2
**Presentation:** 3
**Contribution:** 3
**Rating:** 2
**Confidence:** 4

**Summary:**

This work proposes a parameter-free approach for promoting equivariance in neural networks by imposing, through an additional loss term, that the intermediate latent representation of the network contains multiples of the regular representations. The authors' main theoretical argument motivating this choice is that, for an injective function equivariant to a finite group, if its codomain is sufficiently large, then it almost surely contains the regular representation. The authors empirically evaluate this claim by first testing whether simple networks trained to be equivariant indeed learn latent representations that contain the regular representation. Additionally, they showcase across various approximate equivariant tasks that promoting this structure in the latent space of an encoder improves performance compared to previous approximate equivariant methods, which typically achieve equivariance through specific network design.

**Strengths:**

- The simplicity of the proposed method allows it to be clearly presented and easily implemented, which may encourage broader adoption, especially compared to more complex works on approximate equivariant architecture.
- The experimental results provide empirical evidence that unconstrained networks tend to learn latent spaces containing the regular representations, and that explicitly promoting this structure through a loss can improve overall performance. Although, as stated in the weaknesses, these results are limited to relatively small finite groups, the method can still be of interest to the community and may be applicable to a broader range of tasks.

**Weaknesses:**

- The main weakness of this work is the inconsistencies in the proof of Theorem 1, which serves as the motivating result for the method. In the proof, the authors assume an injective equivariant function $E$ and construct a linear map $\tilde{E}$, represented in matrix form as $M$. They argue that if $\det(M)\not =0$ then the output of the equivariant function contains the regular representation, and that $\det(M)\not=0$ almost surely, since random matrices are almost surely invertible. However, this probabilistic argument is problematic, since $M$ is not a randomly sampled matrix. On the contrary, $M$ is constructed from an optimized/trained function $E$. Thus, the claim that $\det(M)\not=0$ almost surely lacks formal justification and doesn't hold in the current version of the proof. There may exist deeper conditions under which $M$ is generically invertible, but these are not discussed in the current proof.
- The experimental results are limited to small finite groups,  such as $C_4$ and $D_3$. In most cases, the more challenging settings for equivariant and approximately equivariant networks involve larger or continuous groups. There is no clear indication whether the results presented in this paper can scale as the dimensionality of the group increases, which will also result in an increase in the latent representation's dimensionality and potentially its computational cost.

**Questions:**

- How can we make a probabilistic argument about the determinant of $M$ and, more broadly, about the nature of the representation of the encoder, when $M$ is constructed by the learned network $E$ and not randomly sampled? Is there any other connection that makes $M$ almost surely invertible?
- How does the method scale for larger finite groups where the size of the latent representation also increases? Did the authors observe any tradeoff, or does the regular representation empirically always seem to be the best choice?

---

> ### Author Response · Authors · 2025-11-20
> **Response to the reviewer (1/2)**
>
> We are grateful for the close attention the reviewer has given to our work, and for these thoughtful comments. We are pleased that the reviewer appreciates the potential of our method for broad applicability. Regarding the reviewer’s questions, we are happy to answer as follows. We have uploaded a revised submission, with changes marked in blue.
>
> • _How can we make a probabilistic argument about the determinant of $M$ and, more broadly, about the nature of the representation of the encoder, when $M$ is constructed by the learned network $E$ and not randomly sampled? Is there any other connection that makes $M$ almost surely invertible?_
>
> Yes, there is. We have comprehensively rigorized the proof of Theorem 1, splitting it into two statements (Theorem 1 and 2) and applying results from measure theory of real analytic functions. We now tie the matrix M to the learned network E, as pointed out by the reviewer. In particular, we show that for an analytic encoder, the set of parameters for which $\det M = 0$ has measure zero, even after finitely many steps of gradient descent. It follows that $M$ is invertible with probability 1, unless a global rank-deficiency property holds which can be easily checked on any orbit. [Theorem 2, and Appendix E]
>
> We also include a discussion on the real analyticity condition in Section 4.2 [Non-analytic encoders paragraph] and Appendix E.1, arguing that this condition is not particularly restrictive. Furthermore, we now empirically verify that the conclusions of Section 4 extend to non-analytic encoders by repeating the exploratory experiments, showing similar results (Appendix F.3).
>
> •  _The experimental results are limited to small finite groups, such as $C_4$ and $D_3$. In most cases, the more challenging settings for equivariant and approximately equivariant networks involve larger or continuous groups._
>
> The finite groups that we study in our work are very prominent and relevant in the equivariant ML/GDL community, and a standard benchmark for equivariant methodologies, including those for infinite groups. We report a non-exhaustive table containing recent literature, peer-reviewed at top-tier ML venues, containing experiments with the groups we considered in our work:
>
> $C_2$
> Weiler & Cesa, NeurIPS 2019, https://arxiv.org/abs/1911.08251
> Finzi et al., NeurIPS 2021, https://arxiv.org/abs/2112.01388
> Maile et al., ICLR 2023, https://arxiv.org/abs/2210.05484
> Zhao et al., ICLR 2023, https://arxiv.org/abs/2210.05484
> Wu et al., ICLR 2025, https://arxiv.org/abs/2408.11760
> Wang et al., ICML 2025, https://arxiv.org/abs/2310.02299
> Veefkind and Cesa, ICML 2024, https://arxiv.org/abs/2406.03946
>
> $C_4$
> Cohen & Welling, ICML 2016, https://arxiv.org/abs/1602.07576
> Weiler & Cesa, NeurIPS 2019, https://arxiv.org/abs/1911.08251
> Finzi et al., NeurIPS 2021, https://arxiv.org/abs/2112.01388
> Wang et al., ICML 2022, https://arxiv.org/abs/2201.11969
> Maile et al., ICLR 2023, https://arxiv.org/abs/2210.05484
> Zhao et al., ICLR 2023, https://arxiv.org/abs/2210.05484
> Wu et al., ICLR 2025, https://arxiv.org/abs/2408.11760
> Wang et al., ICML 2025, https://arxiv.org/abs/2310.02299
> Veefkind and Cesa, ICML 2024, https://arxiv.org/abs/2406.03946
> Pertigkiozoglou et al., NeurIPS 2024, https://arxiv.org/abs/2408.13242
>
> $D_4$
> Cohen & Welling, ICLR 2017, https://arxiv.org/abs/1612.08498
> Weiler & Cesa, NeurIPS 2019, https://arxiv.org/abs/1911.08251
> Finzi et al., NeurIPS 2021, https://arxiv.org/abs/2112.01388
> Zhao et al., ICLR 2023, https://arxiv.org/abs/2210.05484
> Veefkind and Cesa, ICML 2024, https://arxiv.org/abs/2406.03946
> Zhang et al., NeurIPS 2024, https://openreview.net/pdf?id=HRnSVflpgt
>
>
> We would also like to emphasise that we have extensively benchmarked our method against continuous groups, namely SO(3), O(3) and the conformal group, handling these cases by choosing appropriate large finite subgroups, i.e. $\text{Sym}_\text{Cube}$ (order 24) and $O_h$ (order 48). We show in the majority of these cases that our method is comparable with or outperforms other heavily-parameterised methods that are specialised to continuous groups (Tables 4 and 5(b)).

---

> > ### Author Response · Authors · 2025-11-20
> > **Response to the reviewer (2/2)**
> >
> > • _There is no clear indication whether the results presented in this paper can scale as the dimensionality of the group increases, which will also result in an increase in the latent representation's dimensionality and potentially its computational cost. … How does the method scale for larger finite groups?_
> >
> > Our results show that the method scales well with group size, and we observe a positive correlation between performance and group size. For example, in the DDMNIST experiment, the largest gain over baseline is obtained with $D_4$, which is the largest group considered in this sequence of experiments, and our method is the best performer for the groups $\text{Sym}_\text{Cube}$ (order 24) and $O_h$ (order 48), even when compared to models that are (approximately) equivariant to continuous groups (O(3), SO(3) and the conformal group, see Tables 4 and 5(b)).
> >
> >
> > We thank the reviewer again for their careful comments and look forward to their response.

---

### Meta-Review · Area_Chair_Za1t · 2026-01-07

**Summary:**

The submission proposes a parameter-free method to enforce approximate equivariance in neural networks by introducing an auxiliary loss that encourages the latent space to contain copies of the regular representation of the symmetry group. The theoretical motivation is based on representation theory, with proofs suggesting that equivariant encoders almost surely include the regular representation under certain conditions. Authors have added empirical evaluations across multiple datasets and groups (including finite and approximated continuous groups) show competitive or superior performance compared to specialised equivariant architectures, while maintaining low parameter counts.

Overall, the reviewers have been critical and quite consistent among them, raising concerns about several issues such as: The probabilistic argument in Theorem 1, the measure-theoretic basis for “almost surely”, the significance of Theorem 2 and its practical relevance, Ambiguity around terms like “linearly independent orbit”, and overall the novelty and rigour.

**Reviewer Concerns:**

The concerns mentioned earlier remained unresolved even after the author rebuttal. There were actually good discussions between the authors and reviewers, but they did not resolve the concerns or indicate that scores would increase.

**Reviewer Scores:**

The reviewers would not increase the score if they had the chance. This is clear given the discussions that took place.

---

### Decision · Program_Chairs · 2026-01-26

Reject